# Functional imaging of visual cortical layers and subplate in awake mice with optimized three-photon microscopy

Murat Yildirim [1,2], Hiroki Sugihara [1], Peter T.C. So [2,3] & Mriganka Sur [1,4]

Two-photon microscopy is used to image neuronal activity, but has severe limitations for studying deeper cortical layers. Here, we developed a custom three-photon microscope optimized to image a vertical column of the cerebral cortex > 1 mm in depth in awake mice with low (<20 mW) average laser power. Our measurements of physiological responses and tissue-damage thresholds define pulse parameters and safety limits for damage-free three-photon imaging. We image functional visual responses of neurons expressing GCaMP6s across all layers of the primary visual cortex (V1) and in the subplate. These recordings reveal diverse visual selectivity in deep layers: layer 5 neurons are more broadly tuned to visual stimuli, whereas mean orientation selectivity of layer 6 neurons is slightly sharper, compared to neurons in other layers. Subplate neurons, located in the white matter below cortical layer 6 and characterized here for the first time, show low visual responsivity and broad orientation selectivity.

[1] Picower Institute for Learning and Memory, Massachusetts Institute of Technology, Cambridge, MA 02139, USA. [2] Department of Biological Engineering, Massachusetts Institute of Technology, Cambridge, MA 02139, USA. [3] Department of Mechanical Engineering, Massachusetts Institute of Technology, Cambridge, MA 02139, USA. [4] Department of Brain and Cognitive Sciences, Massachusetts Institute of Technology, Cambridge, MA 02139, USA. Correspondence and requests for materials should be addressed to P.T.C.S. (email: ptso@mit.edu) or to M.S. (email: msur@mit.edu)

Multiphoton microscopy has been used to perform high-resolution structural and functional brain imaging in awake and anesthetized mice[1–3]. Unfortunately, two-photon microscopy still has several limitations when it comes to deep brain imaging, including potential in-focus cell damage and phototoxicity due to high optical intensities as well as low signal to background ratio[4,5]. Crucially, with increasing imaging depth, the relative strengths of out-of-focus (background) fluorescence signal relative to signal from the focal region becomes higher due to scattering[6,7]. Thus, the maximum imaging depth reaches a fundamental limit when the background signal generated close to the surface equals the signal arising from the focal volume. This maximum imaging depth is dependent on the tissue's optical properties[7–9]; absorption and scattering due to densely packed neuronal and non-neuronal cells and cellular processes of the cortex limit the imaging depth to 2–3 scattering lengths for green fluorescent protein (GFP) or green genetically encoded calcium indicator (GCaMP) imaging[10,11]. Thus, two-photon microscopy is typically limited to imaging the superficial 300–400 μm of cortex, which spans a depth of 1 mm or more in mice. To overcome this limitation for imaging deep layers of the brain with two-photon microscopy, wavefront correction was applied to image deep layers[12]. Also, confined expression of fluorophore was achieved via either using layer-specific Cre-dependent animals[13], or virus injection[14].

To image all six layers of the cortex with multiphoton microscopy, increasing the excitation wavelength and utilizing three-photon microscopy is reasonable due to increased scattering length at longer wavelengths and reduced out-of-focus background signal[15]. Indeed, such implementation has been shown to improve the maximum imaging depth in brain tissues for structural imaging[15,16]. Recently, in vivo three-photon microscopy at 1300 nm wavelength has been used to record spontaneous activity of hippocampal neurons with GCaMP6s calcium indicator in anesthetized mice[17]. Although this study has shown proof-of-concept of functional deep tissue imaging, assessment of different kinds of tissue damage and functional changes in neurons is a major concern in three-photon microscopy. Since multiple studies demonstrate that increase in temperature may result in physiological changes in the brain without any visible damage[18,19], optimization of laser and microscope parameters through characterization of optical properties of the brain is required to enable valid, reliable and damage-free recording of evoked neuronal responses in deep brain regions in awake mice.

Consideration of laser parameters such as pulse width and repetition rate, along with optical parameters such as scan lens, tube lens, objective, and collection optics, is crucial for developing an energy-efficient microscope for reducing or eliminating tissue damage in physiological imaging experiments. The number of photons absorbed per fluorophore in three-photon excitation is inversely proportional to the square of the pulse width times the repetition rate (see "Methods" section for detailed explanation). Thus, reducing the pulse width and/or repetition rate automatically increases the number of photons absorbed per fluorophore. However, decreasing the repetition rate results in low imaging frame rate. Furthermore, lowering the pulse width to very short pulses may induce higher-order dispersion, which may elongate the pulse width on the sample. Design parameters for the microscope can be categorized into two groups, as ones related to the excitation path and the emission path. Excitation path parameters consist of shaping the laser beam corresponding to the scanning mirrors, designing the scan and tube lens to reduce aberration in the system, and designing the objective lens to increase transmission of the laser beam with uniform illumination. On the other hand, optimizing the emission path consists of designing an objective lens to increase the transmission of emitted

light from the brain, and collection optics to increase the signal acquired for different imaging modalities. Although there has been some effort to maximize the efficiency of two-photon fluorescence microscopy[20,21], there has been no similar analysis for three-photon microscopy. Importantly, better characterization of the optical properties of the live brain is required to select parameters for three-photon microscope design. The method used in recent studies is based on labeling blood vessels with high two-photon absorption cross-section dyes and imaging them with two-photon microscopy at 1300 nm[15,22]. Alternatively, label-free imaging of blood vessels with third harmonic generation (THG) imaging can also provide information about scattering properties of the brain at 1300 nm wavelength, since there are well-known Soret transition peaks of oxy- and deoxyhemoglobin at 415 and 430 nm, respectively[23]. Both of these methods rely on the assumption that ballistic photons are mostly responsible for the signal acquired in the photon multiplier tube (PMT). In order to determine the optical properties at the excitation wavelength more precisely, creating small laser ablation spots has been suggested for turbid tissues[9,24]. In these studies, tissues are ablated at different depths with different pulse energies to determine the tissue optical properties and threshold energies for ablation in each tissue layer.

Here we have used the above measurements and demonstrate the development of an energy-efficient, damage-free three-photon microscope, utilizing optical properties of the live mouse brain. We have measured visual responses of neurons across cortical layers of the primary visual cortex (V1) in mice. Importantly, we have imaged responses of neurons in the deep layers of V1 and, for the first time, in the subplate, in the white matter below cortical layer 6, and demonstrated their unique response properties. These measurements demonstrate the feasibility of optimized three-photon microscopy for live deep-tissue functional imaging at subcellular resolution.

## Results

**Optimizing laser and microscope parameters**. We constructed a three-photon microscope (Fig. 1a; see "Methods" section) and focused on optimizing laser parameters to image GFP and GCaMP6s labeled neurons in awake mice with minimal pulse energy. The wavelength of the noncollinear optical parametric amplifier (NOPA) system varied between 1280–1350 nm with 10 nm intervals. The excitation wavelength was optimized at 1300 nm to maximize both GFP and GCaMP6s signals. Since the optics inside the microscope elongated the pulse width on the sample, a two-prism based pre-chirp system was developed to compensate for pulse broadening so that the pulse width on the sample was reduced to 40 fs. This short pulse width can allow us to perform three-photon imaging of all cortical layers of a mouse brain with remarkably low (<20 mW) average power (see "Methods" section and below). Furthermore, a delay line (Supplementary Figure. 1) was built to double the repetition rate for obtaining a reasonable frame rate for calcium imaging (Fig. 1a).

We designed the intermediate optics in the excitation path to maximize the efficiency of GCaMP6s and GFP excitation. Since there are no exact specifications of individual elements of a commercial objective (XLPN25XWMP2, Olympus) that we used in this study, we modeled the objective in Zemax software via iterative optimization using a published Olympus objective design as the starting point[25]. The objective design (Fig. 1b) was optimized for overall transmission, clear aperture, overall length, effective focal length, working distance, and lateral resolution at 1300 nm with maximum acceptance angle of 2°. The point spread function (PSF) of each acceptance angle (Fig. 1c) and calculated spot sizes in this objective design corresponded to an effective

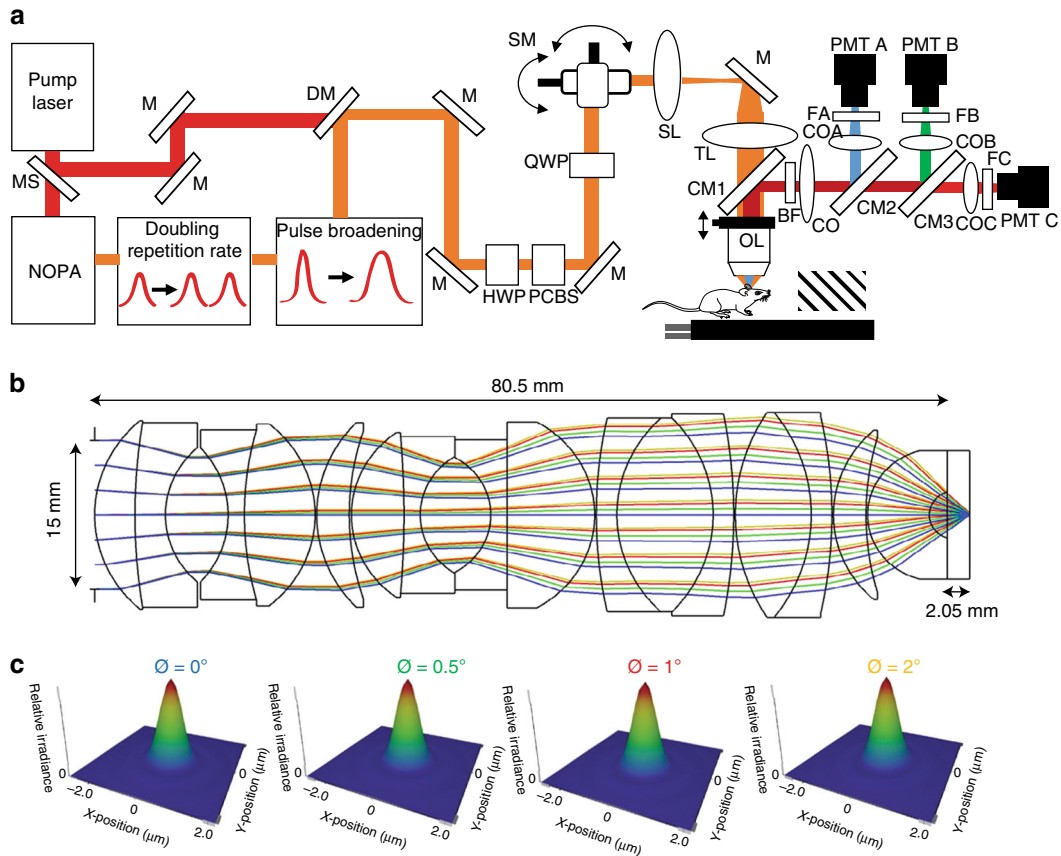

**Fig. 1** Schematic of upright three-photon microscope and layout of the modeled objective lens. **a** Femtosecond laser pulses from the pump laser (1045 nm) were used to pump the noncollinear optical parametric amplifier (NOPA) to obtain 1300 nm excitation wavelength. The pulse width was elongated using a two-prism pair system and the repetition rate was doubled to 800 kHz by adding a delay line. Power control of laser pulses was performed using a combination of a half-wave plate (HWP) and polarizing cube beam splitter (PCBS). A quarter-wave plate (QWP) was used to control the polarization state of the laser pulses to maximize third harmonic generation (THG) signal on the mouse brain. Laser beams were scanned by a pair of galvanometric scanning mirrors (SM), which were then imaged on the back aperture of a 1.05 NA, 25× objective lens (OL) by a pair of a scan lenses (SL) and a tube lens (TL). Mice were placed on a two-axis motorized stage. The objective was placed on a one-axis motorized stage. Emitted light was collected by a dichroic mirror (CM1, FF705-Di01,Semrock; CM2, Di02-R488, Semrock; CM3, FF555-Di03, Semrock), collection optics (CO), laser blocking filters (BF, FF01-670/SP), and nonlinear imaging filters (FA, FF01-433/24-25, Semrock; FB, FF03-525/50, Semrock; FC, FF01-630/92) and corresponding collection optics (COA, COB, and COC) for each photomultiplier tube (PMT A, PMT B, and PMT C). To make laser ablation marks at different depths of cortex, the output of the pump laser (1045 nm) was sent to the microscope using a mechanical shutter (MS) and a long pass dichroic mirror (DM, Di02-R1064, Semrock). **b** The optical layout of the objective modeled in Zemax at 1300 nm. The clear aperture of the objective was 15 mm and the working distance was 2.05 mm while using seawater as a sample. **c** Huygens point spread function results for 4 different scanning angles (0°, 0.5°, 1°, and 2°) on the objective. Numerical apertures in all cases were 1.02 and Strehl ratio values decreased from 0.997 to 0.979 while increasing the scanning angle

numerical aperture (NA) of 1.02. This NA agreed well with the actual NA of 1.05 with 25× magnification objective used (other parameters of the objective are tabulated in Supplementary Table 1). The scan and tube lens combination was modeled in Zemax software with a single galvanometric mirror. The collimated beam was sent to a galvanometric mirror and scanned in the angular range of −10 to 10° (representing an effective range of −2 to 2° at the back aperture of the objective lens). Therefore, the 4f imaging system was designed with a scan lens and a tube lens having focal lengths of 75 and 375 mm, respectively. To optimize the design of these two lenses, the designed objective was also included in the model and the aberration in each component was minimized for the full scanning range (Supplementary Figure. 2a). First, field curvature and f-theta distortion were minimized to 2 μm and 0.5 percent (Supplementary Figure. 2b), respectively. Second, spot sizes at different scanning angles on the focal and afocal planes were minimized (Supplementary Figure. 2c). Third, RMS wavefront error for the entire range of incidence angle at the objective was

optimized (Supplementary Figure. 2d). For values up to 1.5° incidence angle, root mean square (RMS) wavefront error was better than or comparable to the diffraction limited case (dashed line) and started to degrade for angles exceeding 1.5°. This objective with higher transmission at 1300 nm wavelength (70%) provided two-fold improvement compared to a similar objective used for two-photon studies (Olympus, 25 ×, 1.05 NA) which has less transmission (30%) at 1300 nm wavelength (Supplementary Figure. 3).

Finally, we designed the collection optics to maximize collection efficiency, taking into account optical properties of the live brain. The emission light path of the microscope was also modeled in Zemax software. The brain tissue was modeled as seawater with corresponding scattering length at each emission wavelength. Previous studies[26,27] reported the average refractive index of cortical neurons as 1.37, which is comparable to that of seawater (refractive index n = 1.34). By using an extinction length of 270 μm (see below), and absorption length of 1 mm for water at 1300 nm[28], the scattering length of brain and anisotropy of

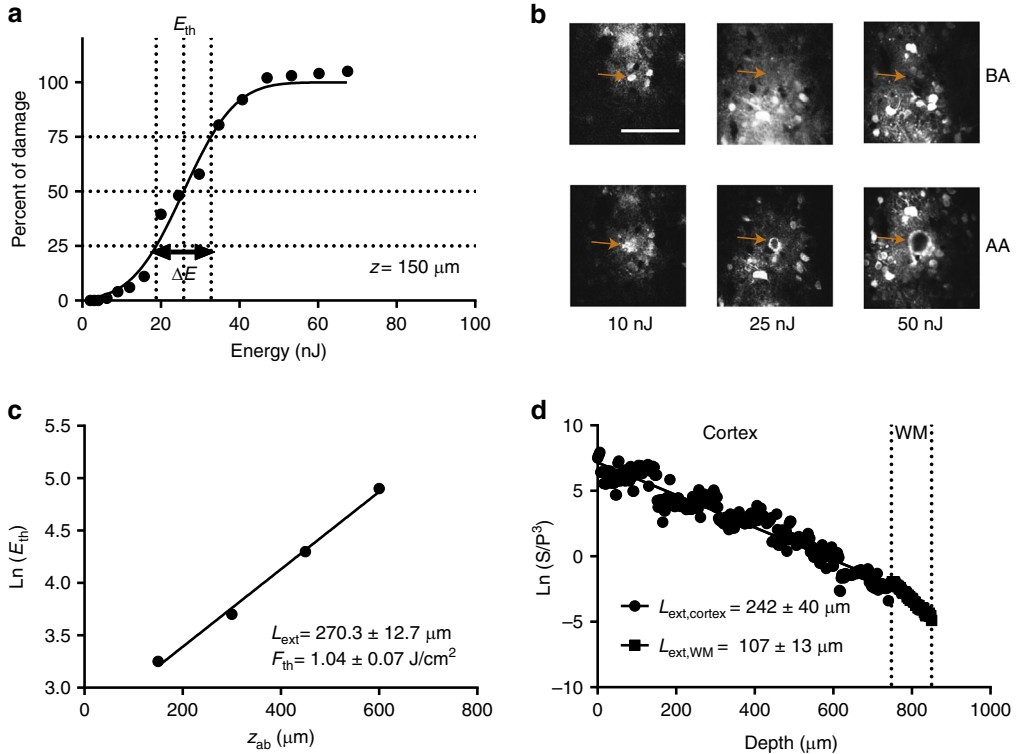

**Fig. 2** Characterization of optical properties of live mouse cortex at 1300 nm excitation wavelength. **a** Determining extinction length via tissue ablation. Percent of damage ranges from 0 to 100% with respect to laser energy on the tissue surface. Threshold energy ($E_{th}$) is the energy which results in 50% damage. For ablation at 150 μm depth, $E_{th}$ is 25.7 nJ. **b** Representative images before ablation (BA) and after ablation (AA) for three pulse energies (10, 25 and 50 nJ) at 150 μm depth. Arrows show the location of target region for the ablation before and after applying laser pulses for each pulse energy. **c** Semi-logarithmic plot of threshold energies for 4 different depths results in extinction length ($L_{ext}$) of 270.3 ± 12.7 μm and threshold fluence ($F_{th}$) of 1.04 ± 0.07 J/cm². **d** Semi-logarithmic plot for ratio of PMT signal ($S$) and cube of laser power ($P$) with respect to imaging depth for third harmonic generation (THG) imaging. Slope of this curve results in 242 ± 40 μm extinction length for the cortex ($L_{ext,cortex}$), and 107 ± 13 μm extinction length for the white matter ($L_{ext,WM}$). Laser power was increased by two-fold every 150 μm in depth. Scale bar in (**b**) represents 100 μm. Values shown are mean ± s.e.m

scattering (g) were calculated as 64, 75, and 90 μm and 0.94, 0.93, and 0.92 for emission wavelengths of 433 nm (Supplementary Figure 2g), 520 nm (Supplementary Figure. 2f), and 600 nm (Supplementary Figure. 2e), respectively by using Mie theory. Thus, for $10^6$ photons generated in the focal plane at 1 mm depth in seawater with bulk Henyey-Greenstein scattering[29], the number of collected photons in each PMT was evaluated with collection optics. With 2-inch dichroic mirrors, a 2-inch collection lens and 1-inch emission lens to optimize collection, the number of collected photons in each PMT was doubled compared to 1-inch collection optics[17] (Supplementary Figure. 4). In addition, we compared these two collection schemes in vivo experimentally by changing the aperture size of an iris placed after the first dichroic (CM1, Fig. 1a) from one inch to two inches (Supplementary Figure. 5a). We found that two-inch optics provides two-fold improvement compared to one-inch size optics while imaging axonal tracts in the white matter via THG microscopy (Supplementary Figures. 5b, c). Overall, optimization of laser parameters and microscope design enabled us to develop a highly efficient three-photon microscope so that we could record neuronal responses through all cortical layers and the subplate of V1.

**Effective attenuation length and damage thresholds**. The effective attenuation length (EAL) comprises of both scattering and absorption length at the corresponding excitation wavelength. EAL of somatosensory cortex in mouse brain has been characterized through two-photon fluorescence microscopy by imaging the vasculature labeled by red dye[15,22]. This method

measured an extinction length of 285 μm at 1280 nm, by calculating the slope of signal attenuation of the two-photon signal with respect to depth. Here, we used two alternative label-free methods to calculate the extinction length in awake mouse brain. The first method was based on creating small lesions at different depths with an excitation wavelength of 1300 nm. To apply this method in awake mouse cortex, four different depths ranging from 150 to 600 μm in 150 μm increments were ablated. A characteristic plot showing percent ablation versus laser energy at brain surface (Fig. 2a) for 150 μm ablation depth shows that 25.7 nJ pulse energy on the surface was required to obtain 50% ablation, where the targeted ablation diameter was 25 μm. To calculate the percent of ablation damage, GCaMP6s images were obtained before and after ablation with varying pulse energies from 10 to 50 nJ (Fig. 2b, see Supplementary Figure. 6 for ablation depths of 450 and 600 μm). The slope of the threshold energy ($E_{th}$) versus ablation depth (Fig. 2c) corresponded to 270.3 ± 12.7 μm extinction length, while the y-axis intercept provided a threshold fluence of 1.04 ± 0.07 J/cm² (mean ± standard error of the mean (s.e.m.), see "Methods" section). This threshold is in good agreement with theoretical estimate of laser-induced optical breakdown in water[30] As described in the previous section, we used this extinction length to optimize the collection optics in the emission path of the microscope. Since only the excitation wavelength is included in these ablation experiments and its analysis (Fig. 2a), this method provides a more precise estimation of the effective attenuation length of the cortex.

The second method of obtaining extinction length was based on calculating the slope of the attenuation signal acquired with

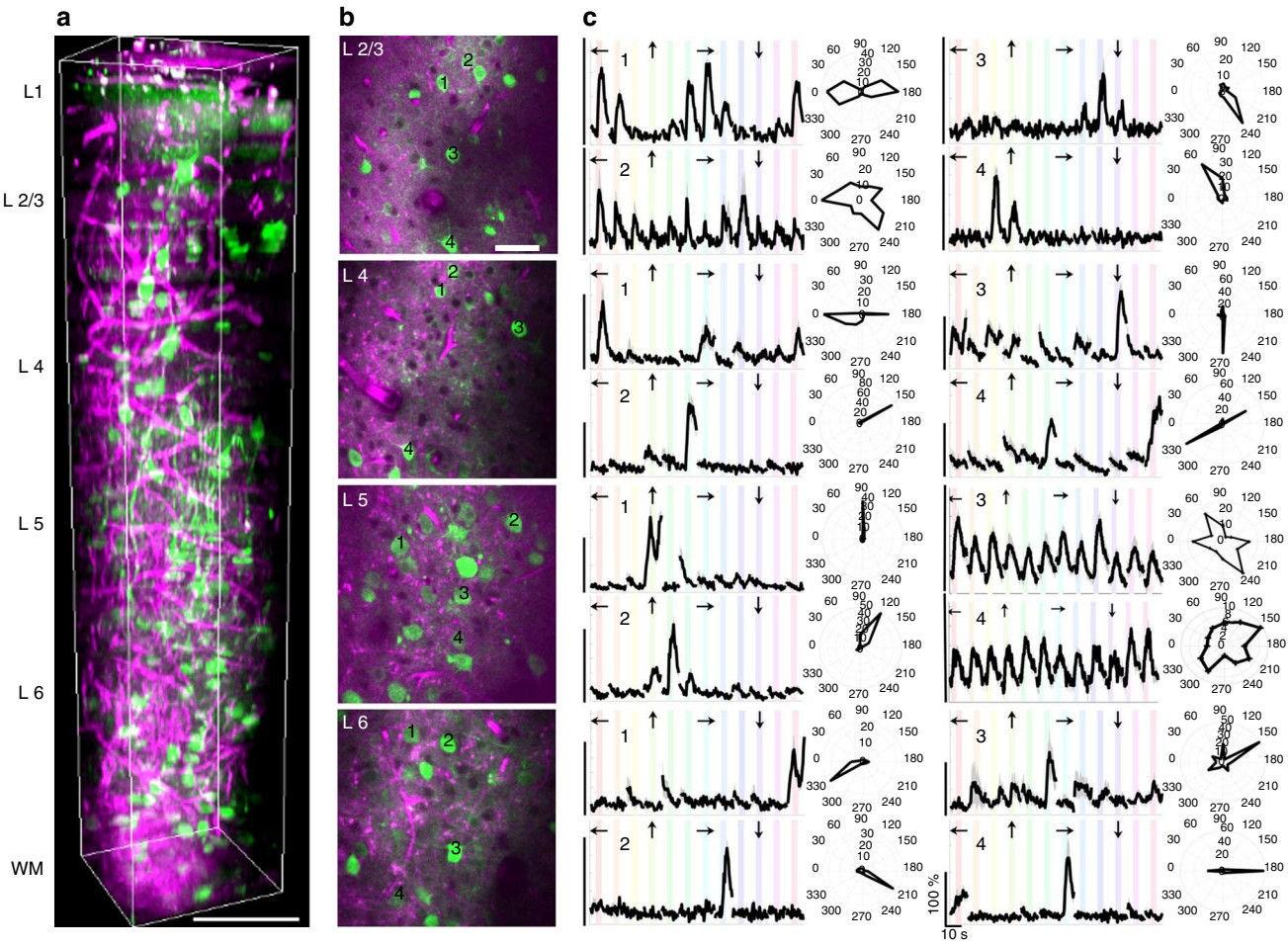

**Fig. 3** Characterization of visual responses of different layers in V1 of awake mice. **a** Three-dimensional rendering of a sequence of 450 lateral three-photon images acquired with 2-μm increment. Green color represents GCaMP6s signal, and magenta color represents label-free THG signal generated in the blood vessels in the visual cortex and myelin fibers in the white matter. Scale bar, 100 μm. **b** Selection of lateral images from layer 2/3, 4, 5, and 6. Scale bar, 50 μm. Field of view in all lateral images is 250 μm. **c** Average calcium responses (ΔF/F) for representative cells in each layer over 10 trials in response to oriented gratings moving in specific directions (arrows above each trace) and their orientation tuning curves in polar plots. Discontinuities in (ΔF/F) traces are due to the randomization of each stimulus direction in each trial. (ΔF/F) scale bars correspond to 100% and time scale bar corresponds to 10 s as shown in the bottom right panel

THG microscopy of blood vessels. Since blood has a one-photon Soret-band transition peak at 430 nm[23], THG imaging at 1300 nm was used to get a reliable signal from blood vessels. In addition, three-dimensional depth-resolved THG image at 1300 nm (Fig. 3a) was similar to the images of labeled blood vessels acquired with two-photon microscopy. Therefore, we conclude that most of the THG signal was generated from the blood vessels. THG signal attenuation curve resulted in 242 ± 40 μm extinction length in the cortex and 107 ± 13 μm extinction length in the white matter, respectively (mean ± s.e.m., Fig. 2d). Thus, THG imaging of blood vessels estimates an extinction length that is comparable to that acquired with ablation experiments.

The threshold fluence for ablation enabled us to estimate safety limits for laser pulse energy to avoid linear and nonlinear absorption-related tissue damage (see "Discussion" section). For example, the threshold energy of optical breakdown (nonlinear absorption) was 130 nJ (~$e^{4.9}$ nJ according to Fig. 2c at 600 μm depth), which is an order of magnitude higher than the required energy (~8 mW /800 kHz = 10 nJ) for three-photon imaging at 600 μm depth. Typical average power for imaging cortical layers ranged from 0.5 to 10 mW. In the white matter, given the higher scattering, typical average power for imaging ranged from 10 to 20 mW. Thus, the three-photon laser parameters necessary to

image the entire cortex are well within safety limits without causing tissue damage.

To further reveal the safety limits for laser pulse energy to avoid any saturation of the GCaMP6s fluorophore or any physiological changes in the neurons, we varied the pulse energies on the focal plane between 0.5 and 15 nJ, and measured evoked responses of neurons across all layers of the visual cortex and in the white matter. We found that it was safe to image with energy pulses in the range of 0.5–5 nJ in our experimental conditions in each layer for avoiding visible damage or saturation of the chromophore (Supplementary Fig. 7). When the laser energy was between 5 and 10 nJ, saturation of GCaMP6s was initiated and the entire field of view was gradually saturated. Finally, optical breakdown was observed when the pulse energy was >10 nJ. Furthermore, we studied the evoked neuronal responses of each layers while the pulse energy was <5 nJ. We classified laser pulse energies into two subcategories as follows: (i) pulse energies between 0.5 and 2 nJ, (ii) pulse energies between 2 and 5 nJ. We compared both preferred orientation and average calcium signal intensity (ΔF/F) in these two pulse energy groups (Supplementary Fig. 8, third and fourth columns). Across all layers, approximately 70% of the visually responsive cells preserved their preferred orientation in both of these energy groups (Supplementary

Fig. 9a). However, the mean response intensity at the preferred orientation was significantly reduced in the second pulse energy group compared to the first in cells with similar preferred orientation (Supplementary Fig. 9b). This result was consistent across all layers. Thus, it is crucial not to exceed the 2 nJ pulse energy threshold for evoked neuronal response studies with three-photon microscopy, with similar laser parameters as used by us. As a control, we also measured preferred orientation and response intensity of layer 2/3 neurons with pulse energy of 0.5 nJ before and after deep layer imaging. This is also one of our motivations for optimizing our microscope-to not damage superficial layers when imaging deep layers. We found that 95% of these neurons preserved their preferred orientations and peak response intensities (within 15 degrees and 10% respectively; Supplementary Figs. 10, 11). This indicates that imaging deep layers did not affect neurons in the superficial layers.

**Application of optimized three-photon microscopy**. Structural and functional imaging of neurons in mouse cortex was carried out in animals with appropriately labeled sets of neurons. GFP-M mice show near-complete labeling of a subset of layer 5 (and layer 2/3) pyramidal neurons; since the basal dendrites of layer 5 pyramidal neurons spread through layer 6 while their apical dendrites span layers 1–5, they are good candidates for demonstrating three-photon microscopy-enabled structural imaging through the entire cerebral cortex. A representative layer 5 pyramidal neuron (Supplementary Figure. 12a) imaged in a GFP-M mouse in vivo, typical of many imaged neurons, indeed shows the structure of nearly the entire neuronal dendritic arbor. By following its somata and apical and basal dendrites, we could reconstruct the neuron and its dendrites (Supplementary Figure. 12b). Our depth-resolved three-dimensional three-photon images (Supplementary Fig. 12a) agreed well with similar neurons identified in histological sections (Supplementary Fig. 12c, Supplementary Table 2). In addition, we measured lateral intensity profiles at planes 50 μm above and below the somata in layer 5 which provides upper bound for the lateral resolution of the system. For example, we found that full-width half maximum (FWHM) of the lateral intensity profiles is ~ 0.5 μm (Supplementary Figure. 13).

For functional imaging of visual responses, the calcium indicator GCaMP6s was virally expressed in a majority of excitatory neurons in all layers of V1 of wild type mice (Fig. 3). Structural imaging of the entire depth of cortex and the white matter performed at 2 μm increments showed neurons (labeled with GCaMP6s) and blood vessels (with THG imaging) throughout the cortex and in the white matter below cortex, where myelinated axons were also identified with THG imaging (Fig. 3a, Supplementary Video 1). Functional imaging of neuronal GCaMP6s responses, starting from 100 μm depth at 100 μm increments at 4 Hz frame rate, with sinusoidal gratings drifting in 12 different directions as visual stimuli, revealed robust visual responses in all layers. Neurons were first classified as visually responsive, and visually responsive neurons were then classified as orientation selective if their tuning curves were well fit to a Gaussian function (see "Methods" section). In specific experiments, laser ablation was performed at different depths in the posterior and anterior parts of the imaging site. The animal was euthanized and the imaging site and corresponding layers located with the help of laser ablation markers (Supplementary Figure. 14). These marker lesions enabled identification of cortical layers and layer-specific visual responses of neurons from in vivo recordings (Fig. 3b), exemplified by responses of selected cells to visual gratings and polar plots of their orientation tuning curves (Fig. 3c).

To characterize the population neuronal responses for each layer in a vertical column, we calculated the global (or vector-averaged) orientation selectivity index (gOSI), local orientation selectivity index (OSI) based on preferred stimulus responses, and direction selectivity index (DSI) (see "Methods" section) across five animals (Fig. 4; Supplementary Table 3). The overall global and local OSI and DSI values in our measurements (Supplementary Table 4) resembled those in previous measurements of neuronal responses recorded with two-photon imaging in the superficial layers of V1[13,31,32]. As a control, we imaged and recorded evoked responses of the same cells in layer 2/3 with conventional two-photon and with our three-photon microscope. We found that preferred orientation and mean peak fluorescence intensity obtained through two-photon and three-photon imaging were matched in most of the neurons (95%) (Supplementary Figures. 15 and 16).

Our three-photon measurements of responses in layers 5 and 6 represent the first large-scale measurements of identified deep layer neurons in any cortical area. Layer 5 neurons had significantly lower global and local OSI as well as DSI values than neurons in any other layer (Supplementary Tables 5, 6 and 7). In other words, layer 5 neurons were less tuned to the visual stimulus than other layers in the visual cortex, which was consistent with earlier findings based on 'blind' electrophysiological recordings[33].

This confirmation also implied that three-photon imaging preserved the cortex in a physiologically normal condition. Furthermore, layer 6 neurons demonstrated significantly higher global OSI values than those of other layers, a finding that may relate to the demonstration that corticothalamic projection neurons of layer 6, identified previously by intracellular labeling and anatomical axon projections, have been described as highly orientation selective[34]. Thus, this first in vivo imaging study to characterize neuronal responses of V1 across all layers has revealed both a broad view of layer-specific responses in the deep layers as well as findings consistent with those estimated from limited electrophysiological recordings.

In addition to characterizing the visual responses of V1 neurons across cortical layers, we examined for the first time visual responses of subplate neurons in the white matter below layer 6 (again using <20 mW average laser power). Subplate neurons are among the earliest born neurons of the cerebral cortex[35], and are considered to be largely transient neurons necessary for the development of cortical connections and circuits[36–38]. In addition, subplate neurons have been implicated in brain abnormalities including epilepsy[39], and neurodevelopmental disorders including autism[40,41] and schizophrenia[42]. The white matter is composed of bundles of myelinated axons, and subplate neurons are surrounded by these axons; with the help of THG imaging of myelinated axons in the white matter, we determined the location of GCaMP6s expressing subplate neurons precisely (Fig. 5a). Their average calcium transients to oriented drifting gratings revealed that many neurons were robustly driven by visual stimuli, and had well-defined orientation tuning curves (Fig. 5b). However, the proportion of visually responsive neurons in the subplate (45%) was significantly lower than the average proportion of visually responsive neurons in other cortical layers (80%). Population responses of subplate neurons across five animals (Fig. 5b; Supplementary Table 3) showed that the global and local OSI, and DSI, of these neurons were significantly lower than these parameters in any of the V1 layers (Supplementary Tables 5–7). Thus, under our experimental

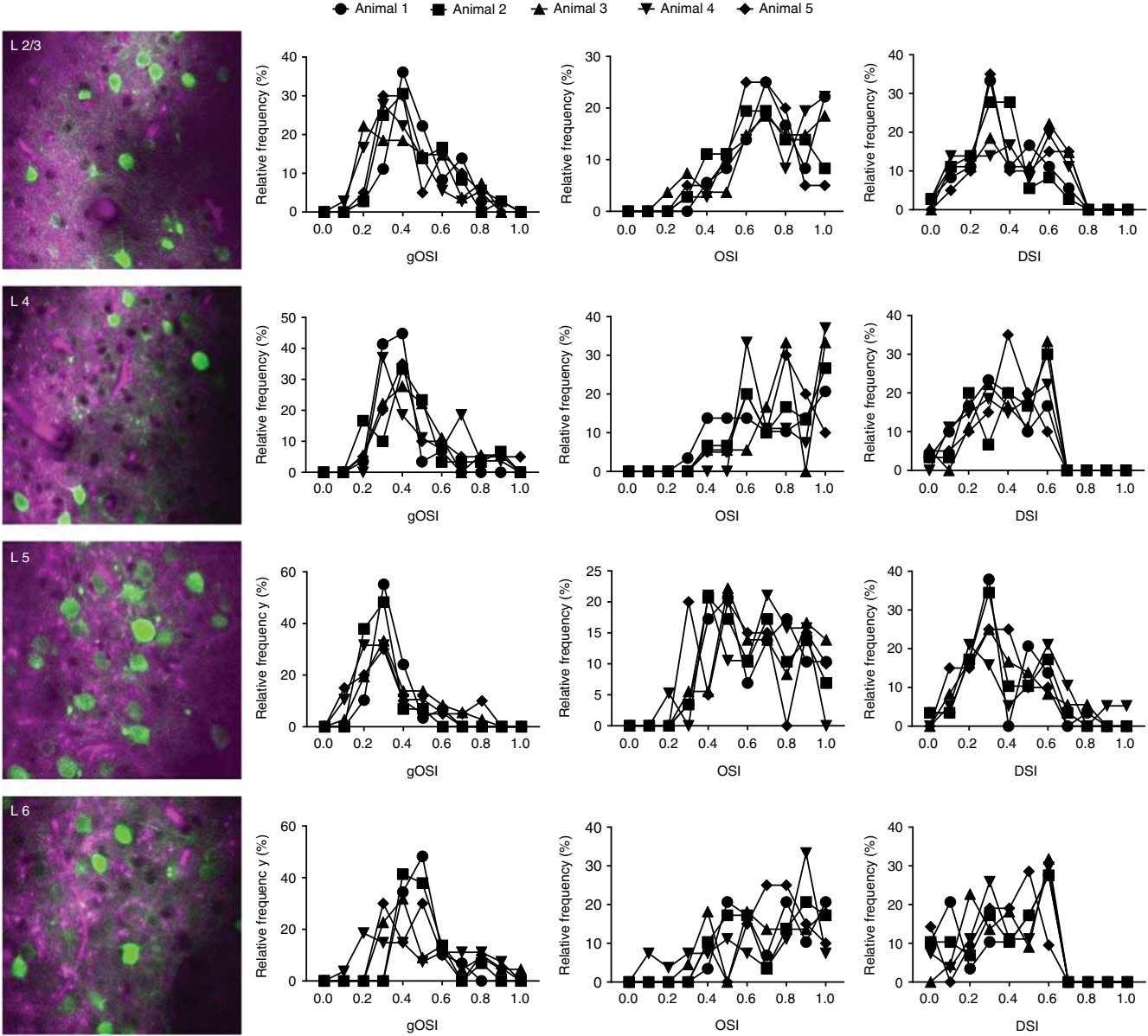

**Fig. 4** Population neuronal responses in each layer across 5 animals. Global orientation selectivity index (gOSI), local orientation selectivity index (OSI), and directional selectivity index (DSI) of neurons in layers 2/3, 4, 5, and 6. Total numbers of orientation selective neurons imaged in each animal are 191, 181, 176, 171, and 155. Their laminar location is listed in Supplementary Table 3

conditions, subplate neurons in V1 are both less visually responsive and less orientation- and direction-selective than neurons within cortical layers. Our study also demonstrates, via imaging that subplate neurons can survive into adulthood and at least some proportion of them receive visual input.

## Discussion
In this study, we focused on optimizing laser and microscope parameters to carry out damage-free measurement of visual responses of neurons in all cortical layers and subplate of the primary visual cortex (V1) of awake mice via three-photon microscopy. We developed a pre-chirp system to compensate for pulse broadening in the microscope and were able to reduce the pulse width to 40 fs on the sample. Furthermore, we implemented a delay line to increase the pulse repetition rate and improve the frame rate. We also designed the optics in the excitation and

emission path to maximize the efficiency of the microscope. In the excitation path, we designed a scan and tube lens integrated with the objective lens to reduce the aberration in the microscope. In addition, we developed two label-free methods to characterize optical properties of the live mouse brain and we used these properties to design the collection optics. These advances resulted in our ability to image GCaMP6s labeled cortical layer neurons with <10 mW and subplate neurons with <20 mW average laser power[17]. Notably, this is nearly an order of magnitude less average power than that required to image layer 5 neurons in layer-specific Cre animals, in which only layer 5 neurons expressed calcium indicator to improve signal-to-noise ratio, via two-photon microscopy with wavefront shaping[13]. Other two-photon studies also similarly utilized an order of magnitude higher laser power to image either layer 5[14] or layer 6 neurons with red-shifted calcium indicators[43]. An order of magnitude less power usage with three-photon microscopy results from several

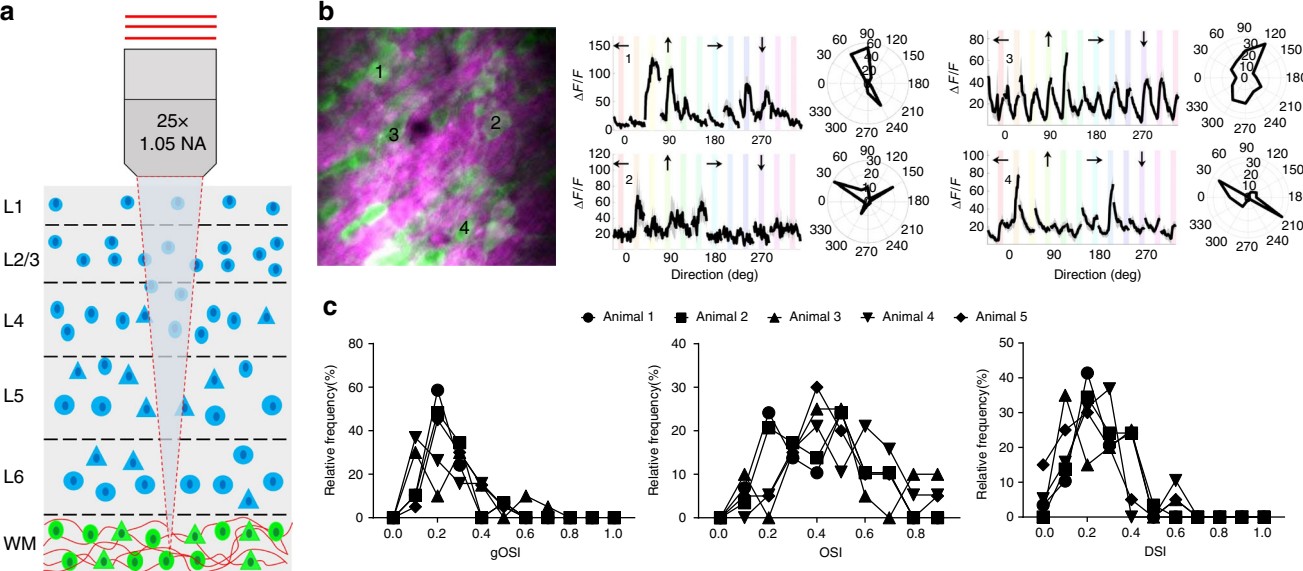

**Fig. 5** Characterization of visual responses of subplate neurons in the white matter below V1 layer 6. **a** Schematic coronal section of V1 showing locations of cortical (blue) and subplate (green) neurons surrounded by bundles of myelinated axons (red). **b** Left: Three-photon image of subplate neurons (green) and myelinated axons (magenta) in the white matter. Scale bar, 50 μm. Right: Average calcium responses ($\Delta F/F$) for representative cells in response to oriented gratings drifting in specific directions (arrows above each trace) and their orientation tuning curves in polar plots. Discontinuities in ($\Delta F/F$) traces are due to the randomization of each stimulus direction in each trial. **c** Population responses of subplate neurons showing global OSI, local OSI, and DSI. Total numbers of subplate neurons imaged in each animal are 43, 48, 38, 35, and 41

parameters such as duty cycle, wavelength, and collection efficiency. For example, three-photon systems have lower duty cycle than two-photon laser systems, which helps reduce average laser power requirement significantly. Low laser power and pulse energy can reduce the temperature at the focal plane which reduces the probability of bulk heating, fluorophore saturation, and optical breakdown. In our study, we found that the probability of saturating GCaMP6s is very low (<10%) when applied pulse energy at the focal plane is 5 nJ assuming that three-photon cross-section of GCaMP6s is on the order of $10^{-82}$ cm$^6$ (s/photon)$^2$ (see "Methods" section). This estimate agrees well with our experimental observation that the saturation of GCaMP6s is initiated with pulse energies > 5 nJ where whole field of view is blurred. More interestingly, we observed abnormalities in the neuronal responses with pulse energies between 2–5 nJ where mean response intensity at the preferred orientation is significantly reduced. Thus, we can conclude that these abnormalities is related to the physiological changes in the neurons rather than saturation of GCaMP6s calcium indicator. Low temperature at the focal plane is always beneficial for the sake of neuronal activity and viability of cells since multiple studies have demonstrated that increase in temperature may result in physiological changes in the brain without any visible damage[18,19]. These optimizations for low-power imaging also enable three-photon systems to perform faster and with wider field. The current lasers available for three-photon microscope have low repetition rate. In future, together with higher repetition rate lasers, optimization for low-power imaging will achieve higher speed imaging, and wider field imaging via multifocal multi-site excitation, without tissue damage.

Understanding mechanism of brain tissue damage with femtosecond pulses is crucial for performing physiological experiments with three-photon microscopy. Focused ultrashort laser pulses can create damage in the brain through either linear or nonlinear absorption mechanisms. Linear absorption processes depend on the average laser power and result in brain damage

through bulk heating[9,44]. The nonlinear process of optical breakdown begins with the generation of free electrons through a combination of multiphoton ionization and band-gap (Zener) tunneling[45,46]. To stay within safe parameters, it is crucial to determine the onset of these mechanisms as well as determine which one dominates under which conditions. Our ablation experiments provide insight into safety limits of laser pulse energy for in vivo imaging of mouse cortex (Fig. 2c). We measured the threshold fluence of optical breakdown, $F_{th}$, as 1.04 ± 0.07 J/cm$^2$, which corresponds to 15 nJ pulse energy at the focal plane (Energy = Fluence x spot area, where spot area is derived from the three-photon point spread function; see Supplementary Figure. 17a and "Methods" section). Therefore, we can calculate the threshold energy of optical breakdown at the surface for any ablation depth. The threshold energy at the surface was ~130 nJ while ablating layer 6 (600 μm depth; see Results). By comparison, the maximum pulse energy at the surface for three-photon imaging at the same depth was 10 nJ (~8 mW/ 800 kHz), corresponding to 1 nJ at the focal plane (depth-resolved attenuation due to Beer-Lambert's law; see Methods) Thus, optical breakdown in layer 6 requires nearly an order of magnitude more pulse energy than three-photon imaging of the same layer under our experimental conditions.

It is important to note that increasing the repetition rate of the laser will be able to allow us to perform higher frame rate imaging with faster calcium indicators such as GCaMP6f. However, we need to take into account increasing probability of bulk heating damage with higher repetition rates. Specifically, we can plot how pulse energy at the surface varies with different laser repetition rates (at pulse width of 40 fs), assuming two different threshold power values: 150 mW average laser power as a lower bound since it is slightly higher than the recently reported maximum average power for damage-free mouse brain imaging[17,47], and 300 mW average power as an upper bound since it is the heat damage threshold for two-photon imaging of live mouse brain at 800, 920, and 1040 nm wavelengths[48] (Supplementary Figure. 19).

This analysis shows that optical breakdown initiates laser-induced damage for moderate laser repetition rates (<1 MHz, 150 mW curve). Bulk heating dominates laser damage for higher repetition rates (>2 MHz, 300 mW curve), and the pulse energy required for imaging is always smaller than both bulk heating and optical breakdown up to roughly 10 MHz repetition rates. In other words, it is safe to increase the repetition rate of the laser up to 10 MHz while keeping the pulse width suitably low (<50 fs). Elongating the pulse width would result in increasing threshold energies for both optical breakdown and three-photon imaging. In three-photon imaging, there is a linear relationship between pulse energy and pulse width since the peak intensity dictates efficiency of imaging. However, the threshold energy for optical breakdown increases at a slow pace ($\sim\tau^{0.4}$) for pulse widths < 1 ps[46]. Thus, pulse widths in the range of 100–500 fs will increase the threshold energy for imaging at a faster pace (shifting the dashed line for imaging up in Supplementary Figure. 19), which would result in increasing the risk of observing laser damage due to either optical breakdown ($R < 1$ MHz) or bulk heating ($R > 2$ MHz) during three-photon imaging.

Our in vivo experiments show that neurons in different cortical layers and in subplate of V1 have distinct visual response properties. Our first key finding is that layer 5 neurons are broadly tuned for orientation compared to neurons in other cortical layers. This result is consistent with previous studies based on electrophysiological recordings from all layers of V1[33], two-photon imaging of all cortical layers except layer 6[13], and two-photon imaging studies focused only on subsets of layer 5 neurons[49,50]. Broader orientation selectivity of neurons in deeper layers, especially in layer 5, was reported also in other species: rat[51], cat[52], and monkey[53,54]. Layer 5 neurons also have highest spontaneous activity among neurons from all cortical layers[33], and this may contribute to their broader tuning property. Other difference between neurons in layer 5 and other cortical layers is the high density of subcortical projection neurons in this layer[49,50]. These neurons may accumulate information from multiple cortical layers and cell types, and thus are broadly tuned for stimulus orientation[55]. Our second key finding is that layer 6 neurons have slightly sharper orientation tuning than neurons in other layers. This is consistent with layer 6 containing substantial numbers of corticothalamic projection neurons which have been described as highly orientation selective[34].

Using in vivo three-photon fluorescence (GCaMP6s) and label-free (THG) imaging, our third key finding is that we image for the first time neuronal responses in the subplate of awake mouse V1. Subplate neurons are known to have a crucial role in the formation of thalamocortical and intracortical connections[35,56,57]. While most subplate neurons seem to be transient and disappear after the cortex has formed, a small population remains, and has been termed persistent subplate cells[58]. Remnant subplate neurons form layer 6b and interstitial white matter neurons[59–61]. So far, subplate neurons in the mature brain have not been studied at all due to the technical challenges of imaging these cells in vivo. An especially interesting finding is that the majority of subplate neurons are poorly responsive visually under our experimental conditions, and the visually driven neurons have lower global OSI, local OSI, and DSI values than neurons in any cortical layer. Interestingly, recent in vitro studies have shown that modulators of arousal, such as neurotensin, strongly depolarize subplate neurons[62,63], suggesting that subplate neuron responses are modulated by internal states. In addition, similar to neurons in superficial layers[64], layer 6b neurons may also be strongly modulated by locomotion[43]. Their broad stimulus selectivity suggests that subplate neurons may in turn influence cortical state-dependent regulation of diverse visual cortex neurons, a suggestion consistent with a role for these neurons in neurodevelopmental and neuropsychiatric conditions. Further studies are required to address this hypothesis.

## Methods

**Experimental setup.** Ultrashort laser pulses (300 fs, 400 kHz, 16 W) at 1045 nm from a pump laser (Spirit, Spectra Physics) were passed through a noncollinear optical parametric amplifier (NOPA, Spectra Physics) to obtain the excitation wavelength of 1300 nm for GFP and GCaMP6s imaging (Fig. 1a). The internal compressor in NOPA could compress the Gaussian pulse width to 35 fs (which was the transform limited pulse width for 70 nm spectral full-width half-maximum bandwidth). Due to the dispersion in the microscope, the Gaussian pulse width on the sample was ~150 fs. In order to shorten the pulse width on the sample, a two-prism based external compressor was built to pre-chirp the pulse before sending it to the microscope, reducing the pulse width to 40 fs on the sample. The prisms were made of SF10, and the distance between the two prisms were 1040 mm. After pulse broadening, a delay line was built to double the repetition rate and increase the frame rate for GCaMP imaging so that frame rate with $256 \times 272$ pixels for $250 \times 250$ μm² field of view can be increased up to 8 Hz. Power control was performed with the combination of half-wave plate (AHWP05M-1600, Thorlabs) and low-GDD ultrafast beamsplitter (UFBS2080, Thorlabs) with 100:1 extinction ratio. The laser beams were scanned by a pair of galvanometric mirrors (6215H, Cambridge Technologies) to image the laser spot on the back aperture of the objective using a pair of custom-designed scan and tube lenses. The emitted signal from the mouse brain was collected by a pair of collection lenses for three PMTs. GCaMP6s, GFP and retrobeads fluorescence signals were detected using GaAsP photomultiplier tubes (H7422A-40, Hamamatsu, Japan); THG signal was detected using bialkali (BA) photomultiplier tube (R7600U-200). The commercially available objective with high transmission at longer wavelength ($25\times$, 1.05-N.A., XLPN25XWMP2, Olympus) was used. Image acquisition was carried out using ScanImage (Vidrio). Imaged cells were located at a depth of 0–1000 μm or more below the pial surface. Laser power ranged from 0.5–16 mW at the sample depending on depth and fluorescence expression levels. A custom stainless steel plate (eMachineShop.com) attached to a black curtain was mounted to the head plate before imaging to prevent light from the visual stimulus monitor from reaching the PMTs. Awake mice and a monitor were placed on a two-axis motorized stage (MMBP, Scientifica) and objective lens was placed on a single axis motorized stage (MMBP, Scientifica) to move in the axial direction. Mice were fixed on the stage with a sample holder and a head mount was placed on top of the head to minimize motion artifacts during imaging. To determine the imaging locations, we created laser marks at different depths in the mouse brain by directing the pump laser to the microscope using a mechanical shutter and a long pass dichroic mirror.

**Effect of laser parameters on absorbed number of photons.** Number of photons absorbed per fluorophore in the focal volume in three-photon excitation can be formulated as follows:

$$n \sim \frac{P^3\delta}{(\tau R)^2}\left(\frac{NA^2}{2hc\lambda}\right)^3, \tag{1}$$

where $n$ is the number of photons absorbed per fluorophore, $P$ is the average laser power, $\delta$ is the three-photon cross-section of the fluorophore, $\tau$ is the pulse width, $R$ is the repetition rate, $NA$ is the numerical aperture, $h$ is the Planck's constant, $c$ is the speed of light, and $\lambda$ is the wavelength of the excitation (1300 nm in this case).

**Effect of pulse energy on GCaMP6s saturation.** The probability of fluorophore excitation per laser pulse ($\text{Pr}_{\text{pulse}}$) should be less than 0.1 ($\text{Pr}_{\text{pulse}} < 0.1$) to avoid excitation saturation and point spread function broadening. $\text{Pr}_{\text{pulse}}$ can be calculated as follows:

$$\text{Pr}_{\text{pulse}} = \frac{n}{R} = \frac{\delta P^3}{\tau^2 R^3}\left(\frac{NA^2}{2hc\lambda}\right)^3 = \frac{\delta E^3}{\tau^2}\left(\frac{NA^2}{2hc\lambda}\right)^3, \tag{2}$$

where $n$ is the number of photons absorbed per fluorophore, three-photon cross-section of the fluorophore[65] ($\delta$) is $1–2 \times 10^{-82}$ cm⁶(s/photon)², average laser power ($P$) is 4 mW, pulse energy ($E$) is 5 nJ, pulse width ($\tau$) is 40 fs, pulse repetition rate ($R$) is 800 kHz, numerical aperture ($NA$) is 0.9, $h$ is Planck's constant, $c$ is speed of light, and wavelength of the laser ($\lambda$) is 1300 nm. With these conditions, $\text{Pr}_{\text{pulse}}$ varies between 0.03–0.06.

**Characterizing microscope performance.** The performance of the three-photon microscope was characterized by measuring the point spread function (PSF), excitation uniformity on the sample, and emission uniformity on the PMTs. Fluorescent beads (100 nm, Invitrogen, F8803) were suspended in agar gel and used to measure lateral and axial resolution at imaging depths ranging from 500 to 1500 μm. The average lateral and axial full-width half-maximum (FWHM) of 20 fluorescent beads were $0.45 \pm 0.05$ μm and $1.9 \pm 0.2$ μm, respectively (mean ± s.e.m., two-dimensional PSF of a representative fluorescent bead is shown in Supplementary Figure. 17a). In addition, we performed in-vivo PSF measurements

specifically in the deep layers of cortex (Supplementary Figure. 18). We injected green retrobeads (Lumafluor) in the visual cortex of one hemisphere and imaged these retrobeads within neurons of L6 extending into L5 and the top of the white matter (WM) in the contralateral hemisphere (which provide callosal projections). We specifically focused on deep layers of the cortex where there is a transition between L6 and WM, so that L5 and 6 can be accurately characterized in vivo. Then, we imaged a column 200 μm high (from below the WM-L6 border through L6 to L5—see Supplementary Figure. 12) at 0.2 μm increments in anesthetized mice. The field of view was 40 μm to image individual beads at high resolution in the lateral direction. The lateral PSF increased from 0.45 to 0.5 μm and the axial PSF increased from 2 to 2.5 μm through L6 to WM. Thus the spherical aberration in the system is fairly low, and degrades the lateral and axial PSF by 10 and 20%, respectively, through L5 and 6 to WM.

To check the excitation uniformity on the sample, PSF analysis was performed for the beads located in four corners and the center of the field of view (Supplementary Figure. 17b, left). This analysis resulted in fairly uniform PSF distribution in the whole field of view (Supplementary Table 8). In addition, a spiky pollen grain was imaged and translated again to the four corners and middle of the field of view while applying the same laser power (Supplementary Figure. 17b, right). The mean intensity and its standard deviation for these five locations are presented in Supplementary Table 9. This analysis also showed a fairly uniform excitation distribution in 250 μm FOV.

To check the emission uniformity on the PMTs, Rhodamine 6 G dye was imaged as a uniform fluorescent dye. This analysis allowed us to correct any misalignment in the collection path of the microscope and determine the effective FOV where the PMT intensity was uniform. The PMT intensity profiles (Supplementary Figure. 17c) show that 1D profile resulted in 250 μm effective FOV with uniform emission from the sample.

**Mice.** All experiments were performed under protocols approved by the Animal Care and Use Committee at the Massachusetts Institute of Technology and conformed to US National Institutes of Health guidelines. In vivo experiments were performed on adult mice between 2–6 months old. Mice of both sexes were used. These animals were housed under 12/12-h light/dark cycle and up to four animals per cage. The following mouse lines were used. Supplementary Figure. 12: Thy1-GFP line M (Stock # 007788, The Jackson Laboratory)[66]. Figures 3, 4 and 5: C57BL6 (wild type, WT).

**Surgical procedures.** Mice were initially anaesthetized with 4% isoflurane in oxygen, and maintained on 1.5–2% isoflurane throughout the surgery. Buprenorphine (1 mg/kg, subcutaneous) and/or meloxicam (1 mg/kg, subcutaneous) was administered preoperatively and every 24 h for 3 days to reduce inflammation. Ophthalmic ointment was used to protect the animal's eyes during the surgery. Body temperature was maintained at 37.5 °C with a heating pad. The scalp overlying the dorsal skull was sanitized and removed. The periosteum was removed with a scalpel and a craniotomy (3–4 mm) was made over the left primary visual cortex (V1, 4.2 mm posterior, 3.5 mm lateral to Bregma), leaving the dura intact. For calcium imaging experiments, neurons were labeled with a genetically-encoded calcium indicator by microinjection (Stoelting) of AAV2/1.Syn.GCaMP6s.WPRE. SV40 (University of Pennsylvania Vector Core). Virus injections were made at a few sites within V1, at depths of 750, 500, and 250 μm below the cortical surface. A volume of 200 nL of virus was injected at 100 nL/min at each depth. After each injection, the pipette was held in place for 2 min prior to retraction to prevent leakage. A circular cover glass (3 mm, Warner Instruments) was implanted over the craniotomy as a cranial window, and sealed with dental acrylic (C&B-Metabond, Parkell) mixed with black ink to reduce light transmission. Finally, a custom-designed stainless steel head plate (eMachineShop.com) was affixed to the skull using dental acrylic. Experiments were performed either at least 5 days after head plate implantation for GFP-M animals, or at least 2 weeks after virus injection for calcium imaging.

**Visual stimulation.** Visual stimuli were generated using custom-written codes with the Psychtoolbox[67] in MATLAB (Mathworks). Oriented grating stimuli were displayed on a 7-inch LCD monitor situated 10 cm from the right eye. Stimuli consisted of full-contrast sine wave gratings (spatial frequency: 0.07 cycles/degree; temporal frequency: 2 Hz) drifting in 12 directions in a pseudorandom sequence for 12 s each. For each trial, a static stimulus was presented for the first and the last 3 s, and the drifting stimulus was presented for the middle 6 s. Ten trials were presented for each stimulus direction.

**Calculation of visual response features.** Neurons were classified as visually responsive when, with at least one of 12 grating directions, a set of 10 responses during drifting grating periods were significantly different from those during static grating periods (paired $t$ test). Orientation selective (OS) neurons were characterized by applying a bimodal Gaussian fit to the tuning curve of each neuron with the following equation:

$$R(\theta) = R_0 + R_p e^{-\frac{\left\langle\theta-\theta_p\right\rangle^2}{2\sigma^2}} + R_n e^{-\frac{\left\langle\theta-\theta_p\right\rangle^2}{2\sigma^2}}, \quad (3)$$

where $\theta$ is the stimulus orientation (between 0 and 360°) and the angle brackets indicate angular values expressed between −180 and 180°. Five parameters were calculated from the above equation: preferred orientation $\theta_p$, tuning width $\sigma$, base response $R_0$, response amplitudes in preferred and opposite directions, $R_p$ and $R_n$, respectively. Cells were classified as orientation selective if the goodness of the fit of the Gaussian ($R^2$) exceeded 0.6[13,68].

Two parameters were used to characterize orientation selectivity of neurons. The first parameter, termed global orientation selectivity index (gOSI), is the vector average of responses of all presented stimulus orientations:

$$g\text{OSI} = \frac{\left|\sum_j R\left(\theta_j\right) e^{i2\theta_j}\right|}{\sum_j R\left(\theta_j\right)}. \quad (4)$$

The second parameter, termed local orientation selectivity index (OSI), is obtained from the responses at the preferred and orthogonal orientations:

$$\text{OSI} = \frac{R_{\text{pref}} - R_{\text{ortho}}}{R_{\text{pref}} + R_{\text{ortho}}}. \quad (5)$$

The directional selectivity index (DSI) was calculated as follows:

$$\text{DSI} = \frac{R_{\text{pref}} - R_{\text{oppo}}}{R_{\text{pref}} + R_{\text{oppo}}}. \quad (6)$$

**Laser ablation experiments.** To determine the optical properties of the visual cortex, laser ablation at 1300 nm at 1 kHz repetition rate was performed. The laser beams were raster scanned at a single depth below the pia surface for the duration of one frame, namely 10 s (0.1 fps), using a pair of galvanometric mirrors. For the targeted ablation field of view of 25 μm and 512 × 512 pixel rate, minimum number of overlapping pulses resulted in 10 s of ablation duration with a 1.3 μm $1/e^2$ one-photon lateral resolution.

Since visual cortex has uniform optical properties in terms of its extinction length, the cortex can be assumed as a single layer tissue so that energy on the surface ($E_{\text{surf}}$) is attenuated by Beer-Lambert's law as follows:

$$F = \frac{E_{\text{surf}} \exp\left(\frac{-z}{l_{\text{ext}}}\right)}{\pi w_0^2}, \quad (7)$$

where $z$ is the ablation depth, $l_{\text{ext}}$ is the extinction length of cortex, $w_0$ is the one-photon $1/e^2$ radius at the focal plane which can be calculated from the three-photon point-spread function[69] (Supplementary Figure 17a), and $F$ is the fluence at the focal plane. To calculate the threshold fluence and extinction length, Eq. 7 can be reorganized as follows:

$$E_{\text{th,surf}} = F_{\text{th}} \pi w_0^2 \exp\left(\frac{z}{l_{\text{ext}}}\right). \quad (8)$$

By taking the natural logarithm of both sides in Eq. 8, a linear regression can be applied to calculate $F_{\text{th}}$ and $l_{\text{ext}}$:

$$\ln\left(E_{\text{th,surf}}\right) = \ln\left(F_{\text{th}} \pi w_0^2\right) + \left(\frac{z}{l_{\text{ext}}}\right). \quad (9)$$

Threshold energy ($E_{\text{th,surf}}$) for each depth is calculated by fitting percent of damage with respect to applied energy on the surface. This fitting function is represented as follows:

$$\text{Damage}(\%) = 50\left[1 + \text{erf}\left(\frac{E - E_{\text{th,surf}}}{1.05\Delta E}\right)\right]. \quad (10)$$

After determining threshold energy for each depth, a linear regression was applied to ablating four different depths in the visual cortex.

To mark visual cortex layers, multiple laser ablation markers were created at multiple depths in a vertical column. The laser ablation markers were created at 250, 500, and 750 μm depths using the pump laser (1045 nm) with the following two sets of pulse energies: (i) 1, 2, and 4 μJ at one location, and (ii) 250, 500 and 1000 nJ at another location, with 20 s laser duration, respectively.

**Image analysis for ablation experiments.** An image analysis algorithm was developed to calculate percent of damage in the GCaMP6s images. First, a median filter with a size of 2 × 2 was applied to smooth the images. Otsu's thresholding method[70] was applied to remove background noise and to convert the images into binary ones. Then, percent of damage was calculated by taking the ratio of difference of total number of pixels below the threshold before and after ablation images to total number of pixels correspond to the targeted ablation area.

**Image analysis for neuronal recordings**. Image analyses were performed with custom written scripts in ImageJ and MATLAB. First, images were corrected for X–Y movement by registration to a reference image (the pixel-wise mean of all frames) using 2-dimensional cross correlation. Neuron cell bodies were manually identified using the reference image in ImageJ. The time lapse change in fluorescence normalized by the baseline fluorescence, $\Delta F/F$, for each neuron was calculated as $\Delta F/F_t = (F_t - F_0) / F_0$, with $F_0$ defined as the mode of the raw fluorescence density distribution.

**Immunohistochemistry**. Animals were deeply anesthetized with 4% isoflurane and perfused transcardially with 0.1 M phosphate-buffered saline (PBS) followed by chilled 4% paraformaldehyde in 0.1 MPBS. The brains were then postfixed in 4% paraformaldehyde in 0.1 M PBS (<4 °C) overnight. The fixed brains were sectioned into 100 μm slices with a vibratome and then blocked in 10% normal goat serum with 0.5% triton in PBS (1 h, 25 °C) before being stained with mouse anti-NeuN (1:250, Millipore, MAB377) overnight (<4 °C). This was followed by a 3 hour incubation in Alexa Fluor 647 goat anti-mouse (1:200, ThermoFisher, A32728) before being mounted on a glass slide with the Vectashield Hardset mounting media with DAPI (Vector Labs). The slides were imaged using a confocal microscope (Leica TCS SP8).

**Statistics**. A normality test was performed to check if samples can be described by a Gaussian distribution before the standard $t$ test was used to compare responses across populations of neurons and across animals. Two-tailed, paired $t$ test was used for comparisons unless indicated. In a small subset of experiments that did not pass the normality test, non-parametric statistics were used. Error bars indicate s.e.m. unless indicated. Blind experiments were not performed in the study but the same criteria were applied to all allocated groups for comparisons. No randomization was performed for the study. No statistical methods were used to pre-determine sample sizes, but our sample sizes are similar to those reported in previous publications.

## Data availability

The data and custom code that support the findings of this study are available from the corresponding authors upon request.

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

## Acknowledgements

This work was supported by US National Institute of Health (NIH) grants EY007023 and NS090473 (MS), 4-P41-EB015871 (PTCS), US National Science Foundation (NSF) grant EF1451125 (MS), a Picower Institute Engineering Collaboration Grant (MS and PTCS; MY and HS), and an equipment grant from the Massachusetts Life Sciences Initiative. We are grateful to Dr. Chris Xu and Dr. Dimitre Ouzounov for critical discussion and technical assistance, and to members of the Sur and So labs for their help.

## Author contributions

M.Y. designed the microscope with input from P.T.C.S. and M.S. M.Y. and H.S performed experiments and analyzed the data. M.Y. wrote the paper and was assisted by H. S, P.T.C.S. and M.S. All authors reviewed and edited the manuscript.

## Additional information

**Competing interests:** The authors declare no competing interests.

