## [Transparent Peer Review File · Nature Communications]

Reviewers' comments:

Reviewer #1 (Remarks to the Author):

The paper by Yildirim et al. describes an imaging technique aiming at visualizing deep cortical or subcortical structures in vivo at high resolution while minimizing potential photo-toxicity. They constructed a microscope that allows three-photon imaging. The authors demonstrate the functionality of this design by providing a proof of principle experiment, imaging visually evoked responses in neurons in all layers of the visual cortex. As expected, this reveals a great diversity of responses.

The paper is divided into two parts: 1) a technical part showing the design of an improved three photon microscope and 2) a biologically functional part showing the orientation tuning of cells in V1 in all layers and subplate.

This is a potentially interesting paper, but it should be noted that it does not represent a unique approach to imaging deep structures. There have been various successful attempts to image deeper layers in the cortex by various means (e.g. adaptive optics, regenerative amplification); and these studies too, have provided proof of principle by showing either population activity of neurons (e.g. Mittmann et al., Nat. Neurosci 2011), and even with high-resolution imaging of neuronal structure (e.g. Wang et al. Nat. Comm 2015). Recently, another study reported a three-photon microscope for deep imaging (Ouzounov et al, Nature Methods, 2017). Possibly due to the lack of comparisons with previous approaches, I was left with some doubts as to how whether this technique provides a true added value apart from the lower powers that were used. That said, if the authors can convincingly demonstrate or argue that this technique provides a strong improvement, I would agree that it could receive broad interest. In addition, some methodological details need to be better described.

Concerns:

1) The exact same technique has recently been published (as mentioned above – Ouzounov et al., Nature Methods (2017) ref 14 of the paper). In a way, the authors themselves underscore this in line 315-316, where they use this reference in the following statement: 'These advances resulted in our ability to image GCaMP6s labeled cortical layer neurons with <10 mW and subplate neurons with <20 mW average laser power'. Nonetheless, the authors claim that the current technique is superior in the sense that less photodamage is induced (line 63-64). However, Ouzounov et al. addressed this issue in their paper, showing that photodamage was negligible (they performed chronic imaging of hippocampal neurons). So, I lost the authors in their argument that the current technique is superior as it potentially produces less damage. My guess is that this entirely based on the finding that they are using less average excitation power. But an actual comparison of inflicted damage is lacking.

2) For characterization of optical properties, the authors do not provide much evidence that they are really recording 'Third Harmonic Generation signals'. There is no description of the detection filters used (I assume that one needs a sharp bandpass filter to prove this).

3) The authors state that their study "reveals that L5 cells are more broadly tuned to visual stimuli whereas L6 are more sharply tuned" (Line 34). However, this conclusion is based on relative population distributions (fig 4), and does ignore the observation that individual neuronal responses are highly diverse and may vary over sub-layers. Especially the L6 distributions look highly variable between animals. The authors should provide a more detailed comparison or nuance these statements.

4) The GCaMP images (fig. 3, 4 and 5) look very saturated, and the nuclei look filled. Can the authors explain how these images were generated, and exclude the possibility that these are 'sick' neurons.

Minor concerns:

- 5) The authors state that they have optimized the setup using a customized objective with GVD pre-compensation. But isn't the objective that is listed commercially available as the Olympus XLPlan 25x 1.05? In addition, pre-chirping units are also routinely available commercially. So, I am not sure that these steps provide a strong technological development.
- 6) It is unclear what we are looking at in fig. 2b. This figure needs more explanation and indicators to point out what the figure needs to show.
- 7) Line 101, neurons are not identified.
- 8) Fig.1 there are no details on the filters, lenses, optics in general that are used. The way to double the rep. rate is not explained.
- 9) Line 159, What is the justification to model tissue based on seawater?
- 10) Line 172, Optical properties of mouse brain cannot, in any way, be characterized by vasculature imaging. The penetration of a laser beam can, but this is not exactly what is written.
- 11) Line 212, what do the authors mean by "agreed well". Do they have any quantification that could support this conclusion?
- 12) Line 317, Which Cre-animal?
- 13) Line 319, could authors explain why three photons microscopy could help to image faster and with wide field?
- 14) Line 365, L6 also projects to subcortical structures
- 15) Line 500 "...laser ablation at 1300 nm" and line 415 it is written that the pump laser is used for ablation (1045 nm). Which one is correct?
- 16) Line 505, The authors wrote: "Since visual cortex has uniform optical properties ..." This is not true.
- 17) Fig.Supp3, Cortical Layers based on NeuN labelling is not clear here.
- 18) Line 696, what does a "similar neuron" mean? Is there any quantification supporting this?

Reviewer #2 (Remarks to the Author):

3 photon imaging is a relatively new technique and this study optimizes and applies this technique to the well-studied mouse visual cortex. In particular the authors optimize a microscope for 2 photon imaging, work to minimize photo damage of this relatively new technique, and perform experiments across the depth of V1. The optical design section of the paper will be valuable to others aiming to implement this technique. The physiological experiments describe a few novel response properties of deep neurons. The limitations of the paper are that it tries to both troubleshoot a new technique as well as gain new information about deep V1 neurons. Unfortunately, this lack of focus limits the numbers of controls performed and compromises depth. More careful exposition and discussion of details as well as discussion of prior work would be beneficial.

Major points:

- 1) The authors systematically optimize their microscope for 3p and are rightfully very concerned about power levels. While damage thresholds tend to be determined histologically, functional changes (e.g. change in spike threshold or synaptic kinetics) could occur due to tissue warming at lower power levels. Has this been assessed?
- 2) The authors design a custom microscope objective. To judge the improvement of this objective it would be nice to see examples and quantification of image quality between image obtained with

- a commercial and this custom objective. Will this objective be available to other investigators?
- 3) The frame rates of the imaging system seem very slow (8Hz) and it would seem that obtaining images throughout the entire column would take a lot of time. Thus, when imaging an entire column the superficial layers are exposed to high powers of light for a long duration. How much cumulative photo damage is induced to the superficial layers during such an imaging experiment?
 - 4) The authors frequently claim that the optical properties of visual cortex are uniform. Is that true? I would assume that blood vessels and neurons have different properties.
 - 5) The paper touts the advantage of 3-photon as deep imaging. However, deep imaging including layer 6 of visual cortex has been possible with the advancement of 2-photon imaging e.g. using RAMM (Mittmann et al. 2011), red-shifted indicators (Birkner et al. 2017). These developments are glossed over. A more careful introduction and discussion of the current state of the art including discussion of prior results in mouse V1 would be beneficial.
 - 6) In evaluating the physiological response properties and layer differences it would be important to know how fluorescence signals relate to the underlying cellular spiking events across layers. While this has been studied in detail for 2p, is the sensitivity the same in 3p? Is GCaMP expression the same in all layers? Is the spike to fluorescence relationship the same for cells in the different layers? Answers to these questions would be important to evaluate the differences in visual responses and tuning across layers.
 - 7) The authors only show little raw neural imaging data and only mean responses (see Fig. 3 and 5). What is the variability of responses in each layer? What was the activity without visual stimulation.
 - 8) The authors image at 8Hz frame rates. It seems that such slow speed would be incompatible with faster dyes such as GCaMP6f. Thus, the utility of this technique to study events on faster time scales seems limited. Can you discuss please? How could one overcome this limitation?
 - 9) In the discussion, the authors compare their power levels with those needed in a deep imaging study with 2p and SLMs and claim a magnitude advantage of 3p imaging. This seems overstated. The use of an AAV here vs. a Cre driver in the prior study could explain part of the difference. Moreover, laser powers needed with red-shifted indicators (63mW), Birkner et al. 2017), were not too dissimilar of those used in the current study. A more thorough discussion of prior work might be warranted.
 - 10) The authors seem to claim that they are the first to image deep layers of V1. Prior 2 photon imaging experiments with red-shifted indicators have shown visual and locomotion induced activity in deep V1 (Birkner et al. 2017). Recordings from deep layers have also been performed in other species. Similarities and differences to the imaging results in mouse should be discussed.
 - 11) The authors speculate about the role of persistent subplate neurons in the discussion. It was my impression that many persistent subplate neurons form layer 6b (Marx et al. 2017, Hoerder-Suabedissen et al. 2018). A more thorough discussion would be beneficial for a general audience.
 - 12) The authors use the enhanced depth penetration to image subplate/layer 6b neurons. The authors cite prior physiological recordings from subplate neurons in auditory cortex (Wess et al. 2017) but claim in multiple places that they are the first to record from layer 6b/subplate neurons. I suggest to rephrase these sections to state this study is the first imaging study of subplate/layer 6b neurons in mouse V1.
 - 13) The authors report that layer 6b neurons seem less driven by visual stimuli than more superficial neurons. What is the underlying mechanism? Is that true in other systems and species? Is it truly a weaker response or lower signal to noise because of scattering? The authors mention in the discussion that layer 6b neurons might be modulated by state. Was vigilance state of the imaged animals constant? In fact, the prior imaging study of deep layers in V1 (Birkner et al. 2017) reported that cells in layer 6 were modulated by locomotion. This should be mentioned.

Reviewer #3 (Remarks to the Author):

This is an excellent manuscript that reports findings of both technical and biological significance. One hopes that supplemental material will be provided that will make it possible for other

laboratories to construct the excellent microscope described in this report. Such an advance, like unique genetic material, sequences, and molecular structure coordinates, should be available to the community once the manuscript describing it is published.

The authors are fair in characterizing the advances they describe here as of great significance for imaging of brain structure and function. Most of my comments below are matters of clarification that can be addressed by changes in the text. There is one major issue that probably requires new experiments that must be addressed before the manuscript is suitable for publication.

My major criticism of the manuscript is that the characterization of the microscope function with non-biological samples may not be the same as in the living brain. See my concern about Fig 5a above. To address this point, the authors should measure the point spread function using single retrogradely transported fluorescent beads at different depths within the tissue. The various layers could be labeled with such beads by injecting projection sites of cells in the different layers. This would of course be of most interest in layers 5 and 6, which, conveniently, have distinct projection targets.

Comments below are preceded by line numbers:

149-150: "laser blocking filters (BF), and nonlinear imaging filters (F)" not in Fig 1A.

152: "mechanical shutter (MS) and a short pass dichroic mirror (DM)." not in Fig 1A.

281-219: It would be more informative to see the images of ablations at the deeper sites, 450 or 600 μm in Fig 2b.

208: The apparent point spread function of the soma and basal dendrites in the image of Supplementary Fig 3a does not look very good. Perhaps this is merely because the image in Fig S3a is rendered at very high contrast. One would want to see linear profiles of intensity across the basal dendrites in order to have an accurate idea of the real quality of these images, and to compare it with the point spread functions of Fig 5a. A panel of additional images rendered at lower contrast would also be a worthwhile addition to this manuscript. Perhaps the full z-stack could be provided to the reader for evaluation.

230: The spatial resolution in horizontal by vertical pixels as well as the frame rate should be noted. Was it always 256x256?

241-249: Are the responses in Fig 3C selected single trials or averages of multiple trials? If there were selected, how was the selection done? These cells do not look typical of the layers.

421: Define delta in lines 422-424.

449: Mice have sexes, not "genders"

479: What is meant by "(movement)" in "Neurons were classified as visually (movement) responsive"

479: How many different grating directions of movement were presented? 12? How many response trials were recorded for each stimulus direction?

Table 2 should include the fractions as well as the numbers of orientation selective neurons in each cortical layer in each case.

Tables 4, 5, 6: Are the P-values in these tables corrected for multiple comparisons? It looks like they are not, in which case they are not valid. Please correct.

Reviewer #4 (Remarks to the Author):

This paper leverages the deep penetration capability of long wavelength 3-photon microscopy for imaging the entire depth of the visual cortex and the subplate neurons. While the technology reported here is similar to that in the previous demonstration of 3P activity imaging, this paper focuses on its application. Specifically, the authors carefully measured the ablative damage threshold in the mouse brain and found that the damage threshold is ~ 10x higher than what is required for 3P imaging. This finding is useful for the field. In addition, there are some interesting results on the orientation tuning of L5/6 and subplate neurons, which shows the promise of 3-photon imaging for neuroscience applications. While I recommend publication of this paper overall, there are a number of details that needs to be corrected. In particular, I found that some of the statements in the paper are either exaggerated or misleading.

(1) In the introduction, "...Although this study has shown proof-of-concept of functional deep tissue imaging, a major concern is tissue damage. Optimization of laser and microscope parameters through characterization of optical properties of the brain is required to enable valid, reliable and damage-free recording of evoked neuronal responses in deep brain regions in awake mice." This statement appears to imply that the previous work (ref. 14) was not optimized for damage-free recording, but the range of pulse energy and average power for 3P imaging were discussed in ref. 14, and in fact the pulse energy used in the focal plane in ref. 14 is essentially the same as reported in this paper. Some modification of this statement is probably needed to accurately reflect the existing literature.

(2) "Although there has been some effort to maximize the efficiency of excitation and emission paths of two-photon fluorescence microscopy^{15, 16}, there has been no similar analysis for three-photon." This statement appears to imply that the design for the collection path is different for 2P and 3P imaging. This is not true. The design and analysis are essentially the same for 2P and 3P in terms of signal collection.

(3) "Although this method can label blood vessels and provide information about optical properties of the brain at 1300 nm, it is invasive." It is not clear what the advantages are by using ablation threshold to characterize tissue properties. It is certainly also invasive. While it is valuable to quantify the ablation threshold in brain tissues, I don't think it is a better way to characterize tissue characteristics. In addition, although THG produces stable signal for blood vessels, it is also important to show the signal is not contaminated by other THG sources in Fig.2 (d), since there are structures other than the blood vessel that can also generate THG signal, some much stronger than blood vessels. Therefore, the use of THG to characterize tissue properties is somewhat problematic due to the non-uniform nature of THG signal.

(4) Fig. 2(d). The THG decay curve looked very segmented or stepwise. Not sure if there is any explanation.

(5) Line 99 "...Using half the average power used in a previous design¹⁴," This statement is misleading. While it is true that the power used here is about half, but imaging the hippocampus in ref. 14 requires the penetration of the entire white matter, which necessitates the higher power. Therefore, the comparison in power is not quite valid.

(6) The use of a custom objective lens here is not fully justified. It is somewhat puzzling that they have to re-design an objective with all the parameters similar to commercial versions (e.g., Olympus 25x objective, 1.05NA). How does the custom lens compare with commercial objective lenses? Does it really provide significant performance improvement? If not, it will be misleading for the field to believe that custom objective lenses (probably quite expensive) are necessary for 3P imaging.

(7) It is not clear why field curvature is important in the lens design for in vivo brain imaging.

(8) Line 165-168. It will be much more convincing to compare the collection efficiency experimentally than using design simulations. There were previous reports indicating that large aperture collection path is not necessary, which contradicts the conclusion here. For example, according to Biomedical Optics Express, Vol. 6, pp. 3113-3127, (2015), even at high scattering, 25 to 30 mm aperture optics are sufficient.

(9) The ablation threshold measured here appears to be in good agreement with previous report in PHYSICAL REVIEW B 94, 024113 (2016). Probably useful to cite this paper.

(10) Line 308 "We developed a custom-made pre-chirp system to compensate for pulse broadening in the microscope and were able to reduce the pulse width to 40 fs on the sample. Furthermore, we implemented a custom-made delay line to increase the pulse repetition rate and improve the frame rate." The dispersion compensation and delay line scheme used here are quite standard. Emphasizing "custom-made" is somewhat misleading.

Response to Reviewers

We thank the reviewers for their insightful comments and suggestions. We have performed an extensive series of new experiments that the reviewers suggested or that were necessary to address their comments. The manuscript has been significantly revised to address all comments and concerns. Our point-by-point response is provided below.

Reviewer #1 (Remarks to the Author):

The paper by Yildirim et al. describes an imaging technique aiming at visualizing deep cortical or subcortical structures in vivo at high resolution while minimizing potential photo-toxicity. They constructed a microscope that allows three-photon imaging. The authors demonstrate the functionality of this design by providing a proof of principle experiment, imaging visually evoked responses in neurons in all layers of the visual cortex. As expected, this reveals a great diversity of responses.

The paper is divided into two parts: 1) a technical part showing the design of an improved three photon microscope and 2) a biologically functional part showing the orientation tuning of cells in V1 in all layers and subplate.

This is a potentially interesting paper, but it should be noted that it does not represent a unique approach to imaging deep structures. There have been various successful attempts to image deeper layers in the cortex by various means (e.g. adaptive optics, regenerative amplification); and these studies too, have provided proof of principle by showing either population activity of neurons (e.g. Mittmann et al., Nat. Neurosci 2011), and even with high-resolution imaging of neuronal structure (e.g. Wang et al. Nat. Comm 2015). Recently, another study reported a three-photon microscope for deep imaging (Ouzounov et al, Nature Methods, 2017). Possibly due to the lack of comparisons with previous approaches, I was left with some doubts as to how whether this technique provides a true added value apart from the lower powers that were used. That said, if the authors can convincingly demonstrate or argue that this technique provides a strong improvement, I would agree that it could receive broad interest. In addition, some methodological details need to be better described.

We have added these previous approaches and references for comparison with our study in the Introduction (lines 54-57) and Discussion (lines 373-377).

Concerns:

1) The exact same technique has recently been published (as mentioned above – Ouzounov et al., Nature Methods (2017) ref 14 of the paper). In a way, the authors themselves underscore this in line 315-316, where they use this reference in the following statement: ‘These advances resulted in our ability to image GCaMP6s labeled cortical layer neurons with <10 mW and subplate neurons with <20 mW average laser power’. Nonetheless, the authors claim that the current technique is superior in the sense that less photodamage is induced (line 63-64). However, Ouzounov et al. addressed this issue in their paper, showing that photodamage was negligible (they performed chronic imaging of hippocampal neurons). So, I lost the authors in their argument that the current technique is superior as it potentially produces less

damage. My guess is that this entirely based on the finding that they are using less average excitation power. But an actual comparison of inflicted damage is lacking.

In this manuscript, we claim that we can image the entire visual cortex, through all layers down to the white matter, with much lower laser power (~2 times) by optimizing the three-photon microscope system through its excitation and emission paths. This improvement can enable us to record neuronal responses without altering their physiology. Lower power usage is always beneficial in avoiding perturbation of neuronal activity and viability of cells, since multiple studies demonstrate that increase in temperature may result in physiological changes in the brain without any visible damage¹⁻⁵. In addition, more deleterious physiological damages and other imaging artifacts, such as bulk heating, optical breakdown or fluorophore saturation, may also occur during interactions between femtosecond pulses and the brain. However, since our goal is to monitor neuronal communication, we choose to measure whether pulse energy will affect neuron firing statistics. We believe that firing statistics are more stringent and relevant criteria than other damage/perturbation measures. In these experiments, we applied four different energies (0.5-2, 2-5, 5-10 and >10 nJ) at the focal plane in all layers of the cortex and the subplate while recording evoked neuronal responses of cortical neurons to sinusoidal gratings in awake mice (Supplementary Fig. 7). We found that no physiological change occurred in parameters such as preferred orientation and peak fluorescence intensity with pulse energies less than 2 nJ at the focal plane (Supplementary Figs 8 and 9). However, preferred orientation and peak intensity of cells were significantly changed when pulse energy was between 2-5 nJ (Supplementary Figs. 8 and 9). Further increase in pulse energy (5-10 nJ) resulted more obvious perturbations including fluorophore (GCaMP6s) saturation and then optical breakdown (>10 nJ) in our experimental conditions (Supplementary Fig. 7). While the three-photon study by Ozounov et al.⁶ simply mentioned that there was no visible damage in their recordings, we believe that our detailed analysis of femtosecond laser-brain interactions for three-photon microscopy makes our work distinct, and valuable for the optics and neuroscience fields. We have modified the Introduction for better comparison with the literature (lines 64-67). We have also improved our Results section to include physiological and laser damage experiments (lines 239-262). Finally, we modified our Discussion to clarify the benefit of low power imaging (lines 377-384).

2) For characterization of optical properties, the authors do not provide much evidence that they are really recording 'Third Harmonic Generation signals'. There is no description of the detection filters used (I assume that one needs a sharp bandpass filter to prove this).

We have added the filter specifications (Semrock, FF01-433/24-25) for THG imaging at 1300 nm excitation wavelength and other imaging modalities that we performed in our experiments (lines 134-154; Fig. 1 in the main text). This THG filter has a 24 nm full-width at half maximum (FWHM) spectrum with 433 nm central wavelength, indicating that it is a narrow (sharp) bandwidth filter exactly matching THG emission wavelength at 1300 nm. In addition, we also provide log-log plot of THG signal and imaging power to show that what we acquire from THG signal is a three-photon process (Response Figure 1, below). For a three-photon process, the signal acquired with PMT is proportional to the cube of the intensity of the laser pulses, thus log-log plot should provide a slope of three which is in the range of the slope we found in this experiment.

Response Figure 1. Characterization of the order of nonlinear THG signal acquired in the white matter. (a) Representative THG image of axonal tracts in the white matter, (b) the slope of THG signal intensities versus imaging power verified the third order nonlinear process on a log-log scale plot. Scale bar is 50 μm .

3) The authors state that their study “reveals that L5 cells are more broadly tuned to visual stimuli whereas L6 are more sharply tuned” (Line 34). However, this conclusion is based on relative population distributions (fig 4), and does ignore the observation that individual neuronal responses are highly diverse and may vary over sub-layers. Especially the L6 distributions look highly variable between animals. The authors should provide a more detailed comparison or nuance these statements.

We agree with the reviewer that individual neuronal responses are highly diverse such that neurons in the same layer may range from low to high orientation selectivity index. This is the reason why we focus on population responses in each layer and compare their average responses and standard deviations to draw our conclusions (Supplementary Table 4). As the reviewer suggested, we modified this statement and emphasized that our conclusions are based on average responses of each layer (lines 32-33).

4) The GCaMP images (fig. 3, 4 and 5) look very saturated, and the nuclei look filled. Can the authors explain how these images were generated, and exclude the possibility that these are ‘sick’ neurons.

Some of the neurons have higher fluorescent values in the somata than others, but they are not ‘sick’. In general, ‘sick’ neurons are not as active as ones shown in these figures; these responses are comparable to those of many neurons recorded in previous papers from our lab⁷⁻⁹ and others¹⁰⁻¹².

Minor concerns:

5) The authors state that they have optimized the setup using a customized objective with GVD pre-compensation. But isn’t the objective that is listed commercially available as the Olympus XLPlan 25x 1.05? In addition, pre-chirping units are also routinely available commercially. So, I am not sure that these steps provide a strong technological development.

The objective that we use in this study (XLPlan25XWMP2, 25x, 1.05 NA) is optimized for longer wavelength excitation and available from Olympus. We made a model of the objective in Zemax so that we can simulate intermediate optics for excitation and collection optics for emission side of the microscope to maximize its efficiency. The reviewer is right that pre-chirping units are routinely available commercially; however, there is no other study reporting that they can achieve 40 fs pulse width on the sample with three-photon microscopy. Thus, we believe that not only the GVD system but overall system design enables us to lower the pulse width to 40 fs which is significantly lower compared to other three-photon studies. The conventional 25x Olympus objective used in 2p studies has two main disadvantages compared to the 3p objective that we use in this study. The first disadvantage is overall transmission; the 2p objective has only 30% of transmission at 1300 nm whereas the 3p objective has 70% transmission at the same wavelength. The second disadvantage is that the signal collected with 2p objective is two times less than that obtained with 3p objective under same conditions. In these experiments, we performed THG imaging of axonal tracts in the white matter using both 2p and 3p objectives with average powers of 8 mW (Supplementary Figs. 3a and 3b). We divided each field of view into five sub-regions and calculated the ratio of average signals in these sub-regions acquired by 3p and 2p objectives (Supplementary Fig. 3c). Overall, our 3p objective provides two-fold improvement compared to 2p objective while imaging axonal tracts via THG microscopy. We have modified the text accordingly (lines 162-164).

6) It is unclear what we are looking at in fig. 2b. This figure needs more explanation and indicators to point out what the figure needs to show.

We aim to show damages after applying different pulse energies in Fig. 2b, as described in the main text. We use this information to quantify the extent of damage to calculate the threshold energy for optical breakdown. We have added arrows to point out these regions before and after applying laser pulses, and modified the figure legend.

7) Line 101, neurons are not identified.

We deleted "identified" from the sentence (lines 102-103).

8) Fig. 1 there are no details on the filters, lenses, optics in general that are used. The way to double the rep. rate is not explained.

We have added details of optics in Fig. 1 in the main text, and details of the delay line for doubling the repetition rate in new Supplementary Fig. 1.

9) Line 159, What is the justification to model tissue based on seawater?

The system is modeled as a microscope cover slip having a thickness of 170 μm (BK7, refractive index $n = 1.511$) with seawater ($n = 1.340$) as the sample medium, to account for any spherical aberrations arising from tissue or an optical window. According to previous studies, the average refractive index of cortical neurons is 1.37^{13,14} justifying our assumption of using seawater for modelling the brain tissue. We have added this description (lines 168-169).

10) Line 172, Optical properties of mouse brain cannot, in any way, be characterized by vasculature imaging. The penetration of a laser beam can, but this is not exactly what is written.

We modified the text (see lines 187-192) to emphasize that we can calculate the effective attenuation length (EAL) by vasculature imaging, which is confirmed with ablation experiments. This EAL dictates the penetration of the laser beam. Since this length comprises both scattering and absorption length, it is actually an optical property of the brain.

11) Line 212, what do the authors mean by “agreed well”. Do they have any quantification that could support this conclusion?

We concluded that these neurons were similar in terms of their soma location, specific notch shape of apical dendrites in the superficial layers, shape of basal dendrites, the distance between notch shape apical dendrites to the somata, the distance between the longest basal dendrite and the somata, and somata location. We have now provided these quantifications (Supplementary Fig. 12 and Supplementary Table 2) in the supplementary material.

12) Line 317, Which Cre-animal?

There are multiple studies using layer specific Cre lines to improve signal-to-noise ratio. With these Cre lines, expression of GcaMP6 can be restricted only in one layer; such as Scnn1a-Tg3-Cre, Thy1-GCaMP6 GP4.3, and Rbp4-Cre mice for imaging of L4, L2/3, and L5 neurons, respectively. The strength of 3-photon microscopy is that it does not require layer specific Cre lines, and that it can image deep and superficial layers from the same animals. We have modified the text to clarify this point (lines 373-375).

13) Line 319, could authors explain why three photons microscopy could help to image faster and with wide field?

We try to emphasize that optimization in excitation and emission pathways helps us to maximize the GcaMP6s signal acquired from the mouse brain. Therefore, we manage to minimize the power requirement for GcaMP6s imaging in our experimental conditions. This optimizations will help us when we would like to use higher repetition rate lasers to image higher dynamic calcium indicators such as GCaMP6f to perform single- or multi-site imaging. We have modified the text to clarify this point (lines 384-388).

14) Line 365, L6 also projects to subcortical structures

We modified and improved our discussion accordingly (lines 433-437).

15) Line 500 “...laser ablation at 1300 nm” and line 415 it is written that the pump laser is used for ablation (1045 nm). Which one is correct?

We used two wavelengths, 1300 and 1045 nm for two different purposes. 1300 nm was used to determine optical properties of the cortex, and 1045 nm was used to create laser marks to determine the imaging locations. We have clarified this in the text (lines 488-490).

16) Line 505, The authors wrote: "Since visual cortex has uniform optical properties ..." This is not true.

Optical property of each component of cortex, such as neurons, glia cells, blood vessels, etc., will be heterogeneous. However, optical property of cortex such as effective attenuation length across its depth can be modeled as uniform as long as imaging and ablation experiments show similar results. In our experimental conditions, both ablation and label-free imaging results show similar extinction length over the whole column of visual cortex as shown in Figs. 2c and 2d in the main text. We modified the text to clarify this point (line 590).

17) Fig.Supp3 , Cortical Layers based on NeuN labelling is not clear here.

Now Supplementary Fig. 12: We provided a small field of view of the histology section which is comparable with the field of view of three-photon imaging. We utilized a larger view of the histology section to differentiate individual cortical layers and the white matter according to density of the neurons and their size.

18) Line 696, what does a "similar neuron" mean? Is there any quantification supporting this?

We added a quantification and modified descriptions accordingly. Please also see point 11.

Reviewer #2 (Remarks to the Author):

3 photon imaging is a relatively new technique and this study optimizes and applies this technique to the well-studied mouse visual cortex. In particular the authors optimize a microscope for 2 photon imaging, work to minimize photo damage of this relatively new technique, and perform experiments across the depth of V1. The optical design section of the paper will be valuable to others aiming to implement this technique. The physiological experiments describe a few novel response properties of deep neurons. The limitations of the paper are that it tries to both troubleshoot a new technique as well as gain new information about deep V1 neurons. Unfortunately, this lack of focus limits the numbers of controls performed and compromises depth. More careful exposition and discussion of details as well as discussion of prior work would be beneficial.

We appreciate that the reviewer found the paper valuable.

Major points:

1) The authors systematically optimize their microscope for 3p and are rightfully very concerned about power levels. While damage thresholds tend to be determined histologically, functional changes (e.g. change in spike threshold or synaptic kinetics) could occur due to tissue warming at lower power levels. Has this been assessed?

We agree with the reviewer that, even without observing visible damages, imaging itself can still affect physiological properties of neurons with low laser power. The low power imaging enabled by our optimization will contribute to this point as well. We have performed new experiments to address this issue, to understand the effect of pulse energy on the physiological responses of neurons to sinusoidal gratings. Please see our response to first point of Reviewer 1. Please also see our response to point 3 for similar comparison between measurements before and after deep layer imaging. Most of neurons maintained their preferred orientation and their peak responses were unchanged. We have modified our Introduction and Results to emphasize the importance of low power imaging (lines 64-67), and our new experimental results (lines 239-262). We have also improved the Discussion to clarify the benefit of low power imaging (lines 377-384).

Please also see Reviewer 1, point 1, for technical details related to low power imaging.

2) The authors design a custom microscope objective. To judge the improvement of this objective it would be nice to see examples and quantification of image quality between image obtained with a commercial and this custom objective. Will this objective be available to other investigators?

This objective is available from Olympus (XLPN25XWMP2, 25x, 1.05 NA). We designed intermediate optics for excitation and collection optics for emission side of our microscope by including a model of this objective in Zemax. Please see our response to Reviewer 1's point 5. We have modified the text to clarify this point (lines 162-164).

3) The frame rates of the imaging system seem very slow (8Hz) and it would seem that obtaining images throughout the entire column would take a lot of time. Thus, when imaging an entire column the

superficial layers are exposed to high powers of light for a long duration. How much cumulative photo damage is induced to the superficial layers during such an imaging experiment?

We would like to thank the reviewer for bringing up this point since this scenario is important when imaging deep layers. This is also one of our motivations for optimizing our microscope. We did not observe any photo damage in the superficial layers after imaging deeper layers. To exemplify this case with quantification, we performed evoked neuronal recordings in all layers of the visual cortex starting from superficial layers to the deep layers and recorded the superficial layers again. Cells in layer 2/3 preserved their preferred orientation and mean fluorescence intensity after imaging of deep layers (Supplementary Fig.10). We quantified these two parameters in a population of cells and found that 95 % of the cells preserved their preferred orientation (within 15 degrees; Supplementary Fig. 11A). Similarly, 95% of the cells had similar mean fluorescence intensities in these two conditions (within 10%, Supplementary Fig. 11B; n= 3 animals, 140 cells). We have included this finding in the main text (lines 256-262).

4) The authors frequently claim that the optical properties of visual cortex are uniform. Is that true? I would assume that blood vessels and neurons have different properties.

We agree with the reviewer that each component of cortex will have different optical properties. Our point in this sentence is to claim that effective attenuation length (EAL) as an optical property of cortex across the all layers of the cortex can be modeled as uniform according to our imaging and ablation experiments. In these experiments, we did not see too much difference in the EAL of each layer in the visual cortex (text Fig. 2). However, we observed a remarkable decrease in the attenuation length of white matter (text Fig. 2d) indicating different optical properties from the cortical grey matter⁶. Overall, both imaging and ablation experiments show that the extinction length of the visual cortex can be taken as uniform, which is important to estimate the laser pulse energy/power on the focal plane. We have modified the text to clarify this point (lines 187-192).

Please also see our response to Reviewer 1, point 16.

5) The paper touts the advantage of 3-photon as deep imaging. However, deep imaging including layer 6 of visual cortex has been possible with the advancement of 2-photon imaging e.g. using RAMM (Mittmann et al. 2011), red-shifted indicators (Birkner et al. 2017). These developments are glossed over. A more careful introduction and discussion of the current state of the art including discussion of prior results in mouse V1 would be beneficial.

We thank the reviewer for suggesting these papers for prior literature on deep brain imaging. These studies also exemplify 2p imaging of specific layers (layer 5 or layer 6) with no fluorescence labeling in other layers, and we have modified our Introduction (lines 54-57) and Discussion (lines 373-375) to show how 3p imaging can be advantageous.

6) In evaluating the physiological response properties and layer differences it would be important to know how fluorescence signals relate to the underlying cellular spiking events across layers. While this has been studied in detail for 2p, is the sensitivity the same in 3p? Is GCaMP expression the same in all layers? Is the spike to fluorescence relationship the same for cells in the different layers? Answers to

these questions would be important to evaluate the differences in visual responses and tuning across layers.

We would like to thank the reviewer for bringing up this issue related to characterizing 3p data with cellular spiking events. We performed several experiments and analyses to examine whether our 3p imaging data agrees with 2p imaging, which is used extensively to study cellular spiking events. First, we imaged the same cells in layer 2/3 with both 2p and 3p imaging systems and recorded their evoked neuronal responses (Supplementary Figure 15). Then, we performed population analyses to compare preferred orientations and mean responses (n=3 animals, 150 cells). We found that preferred orientations (95%) and peak responses (94%) obtained through 2P and 3P imaging were matched well in most of neurons (Supplementary Figure 16). This result indicates that relationship of fluorescence signals and spike rates is comparable between 2P and 3P imaging.

In addition, we performed deconvolution analysis of our 3p data imaging spontaneous activities of neurons in all layers of V1 and white matter, by utilizing nonnegative deconvolution (NND)¹⁵ (Response Figure 2). In addition, we plotted spike probability for each layer, normalized by the maximum spike probability (Response Figure 3b). Our analysis showed that L5 has maximum spike probability, followed by L6, L4, white matter and L2/3. Comparing neurons in layers 2-6, our results agree well with electrophysiological recordings in V1 performed by Neil and Stryker¹⁶ for the visual cortex (shown in Response Figure 3a). Thus, in response to the reviewer's question, the spike to fluorescence ratio as imaged with 3p microscopy seems to be well preserved across different layers.

Response Figure 2. Deconvolution of 3p data for each layer. (A) Representative spontaneous single cell 3p recordings at each layer. (B) Deconvolution of 3p recordings of representative cells shown in (A).

Response Figure 3. Comparison of spontaneous spike rate of neurons in different layers of the primary visual cortex. (a) Ground truth electrode recordings obtained by ¹⁶, (b) spike probability of each layer normalized by layer 5 neurons from our 3p imaging data. Statistical significance was determined by Mann-Whitney U test and significant figures are calculated compared to L2-3 (* $p < 0.05$, ** $p < 0.01$, *** $p < 0.001$, and **** $p < 0.0001$). 'Inh': inhibitory neurons inferred from their spike shape. WM, white matter.

7) The authors only show little raw neural imaging data and only mean responses (see Fig. 3 and 5). What is the variability of responses in each layer? What was the activity without visual stimulation?

We have now analyzed the variation of fluorescence signal at the preferred orientation over the course of ten trials (Response Figure 4). This analysis shows that there is an ~20% change in the fluorescence intensity in all layers of the cortex and the white matter. We have also performed new experiments to measure spontaneous activity without visual stimulation (eg., Response Figures 2 and 3). Our 3p imaged responses are consistent with previous reports of spike variability measured with electrophysiological recording (see response to point 6).

Response Figure 4. Percent change of fluorescence intensity of the preferred orientation over the course of ten trials for all neurons recorded in five animals. On average, 20% change occurs for all layer of the visual cortex and the white matter.

8) The authors image at 8Hz frame rates. It seems that such slow speed would be incompatible with faster dyes such as GCaMP6f. Thus, the utility of this technique to study events on faster time scales seems limited. Can you discuss please? How could one overcome this limitation?

One of the bottlenecks for faster 3-photon imaging is repetition rate of the laser. We now discuss the possible options for increasing imaging speed and its limitation (Discussion, lines 407-427). Basically, increasing the repetition rate up to 10 MHz should be safe as long as the pulse width is on the order of 50 fs, which will enable us to image all layers of the cortex with >20 Hz speed at fields of view of a couple of hundred μm in diameter.

9) In the discussion, the authors compare their power levels with those needed in a deep imaging study with 2p and SLMs and claim a magnitude advantage of 3p imaging. This seems overstated. The use of an AAV here vs. a Cre driver in the prior study could explain part of the difference. Moreover, laser powers needed with red-shifted indicators (63mW), Birkner et al. 2017), were not too dissimilar of those used in the current study. A more thorough discussion of prior work might be warranted.

We agree with the reviewer that expression level of GCaMP6 is stronger with viral injection than with transgenic lines. However, layer specific Cre lines also greatly improve signal-to-noise ratio due to limited expression in non-imaged layers (or cells). Furthermore, the power levels with 2p imaging are still an order of magnitude higher than our study in terms of deep layer imaging even with red-shifted indicators that the reviewer suggested (63 mW vs 8 mW). This order of magnitude difference results from multiple factors that affect power requirement for 2p and 3p imaging such as duty cycle (multiplication of pulse width and repetition rate) of the laser. It is obvious that there is approximately two-order of magnitude difference in duty cycle of the lasers used for 2p and 3p systems to get similar signal from the same fluorophore such as GFP (Response Figure 5). Assuming that pulse width for both 2p and 3p microscopy is the same and since pulse energy required for imaging GFP with 2p is approximately order of magnitude smaller than that of 3p microscopy, average power requirement (pulse energy times repetition rate) for 3p microscopy is order of magnitude smaller than that of 2p microscopy. We agree with the reviewer that more discussion is warranted, and have done so (lines 373-388).

Response Figure 5. Comparison of 2p and 3p signal with varying duty cycle of laser systems. Assuming that $\lambda=1300$ nm, NA=1.0, P=5 mW, 2p and 3p cross-sections are 10^{-49} cm⁴(s/photon) and 10^{-82} cm⁶(s/photon)².

10) The authors seem to claim that they are the first to image deep layers of V1. Prior 2 photon imaging experiments with red-shifted indicators have shown visual and locomotion induced activity in deep V1 (Birkner et al. 2017). Recordings from deep layers have also been performed in other species. Similarities and differences to the imaging results in mouse should be discussed.

We have modified the Introduction (lines 102-103) and cited Birkner, A. et al., 2017 (lines 373-375). We also added a comparison of deep layer responses in the primary visual cortex from various species (lines 433-434).

11) The authors speculate about the role of persistent subplate neurons in the discussion. It was my impression that many persistent subplate neurons form layer 6b (Marx et al. 2017, Hoerder-Suabedissen et al. 2018). A more thorough discussion would be beneficial for a general audience.

As the reviewer pointed out, persistent subplate neurons seems to become a part of layer 6b neurons and interstitial white matter neurons (Marx, et al., 2017; Chun and Shatz, 1989; Hoerder-Suabedissen, et al., 2018). In this paper, we imaged activities of subplate neurons which were located below layer 6, where we can observe fiber tracts from THG signal (see Fig. 5B). We have added this to the Discussion (lines 446-447).

12) The authors use the enhanced depth penetration to image subplate/layer 6b neurons. The authors cite prior physiological recordings from subplate neurons in auditory cortex (Wess et. al 2017) but claim

in multiple places that they are the first to record from layer 6b/subplate neurons. I suggest to rephrase these sections to state this study is the first imaging study of subplate/layer 6b neurons in mouse V1.

We have rephrased the statements accordingly (lines 102-103, 438-439).

13) The authors report that layer 6b neurons seem less driven by visual stimuli than more superficial neurons. What is the underlying mechanism? Is that true in other systems and species? Is it truly a weaker response or lower signal to noise because of scattering? The authors mention in the discussion that layer 6b neurons might be modulated by state. Was vigilance state of the imaged animals constant? In fact, the prior imaging study of deep layers in V1 (Birkner et al. 2017) reported that cells in layer 6 were modulated by locomotion. This should be mentioned.

We thank the reviewer for these suggestions. Signal strength of subplate neurons was comparable to those in other layers (please see traces in Supplementary Fig. 8). Also, layer 6 cortical neurons, which are just above the subplate neurons, were more visually responsive. There is little report of sensory responses of adult subplate neurons. Thus, it is an open question if subplate neurons in different area/species have similar response properties to those in mouse V1. As the reviewer pointed out, neurons in deep layers, as well as in superficial layers¹⁷ are modulated by locomotion. Similarly, further studies will be required to see if subplate neurons are differently modulated by locomotion/cortical state than ones in other cortical layers. We have modified our discussion accordingly (lines 453-454).

Reviewer #3 (Remarks to the Author):

This is an excellent manuscript that reports findings of both technical and biological significance. One hopes that supplemental material will be provided that will make it possible for other laboratories to construct the excellent microscope described in this report. Such an advance, like unique genetic material, sequences, and molecular structure coordinates, should be available to the community once the manuscript describing it is published.

We appreciate the generous comments of the reviewer. We have provided more specifications for the microscope (see Methods, Figure 1, Supplementary. Figs. 1, and 2.

The authors are fair in characterizing the advances they describe here as of great significance for imaging of brain structure and function. Most of my comments below are matters of clarification that can be addressed by changes in the text. There is one major issue that probably requires new experiments that must be addressed before the manuscript is suitable for publication.

My major criticism of the manuscript is that the characterization of the microscope function with non-biological samples may not be the same as in the living brain. See my concern about Fig 5a above. To address this point, the authors should measure the point spread function using single retrogradely transported fluorescent beads at different depths within the tissue. The various layers could be labeled with such beads by injecting projection sites of cells in the different layers. This would of course be of most interest in layers 5 and 6, which, conveniently, have distinct projection targets.

Per the reviewer's suggestion, we performed a new experiment for in-vivo characterization of the point spread function in deep layers (Supplementary Fig. 18). We injected green retrobeads (Lumafuor) in the visual cortex of one hemisphere and imaged these retrobeads within neurons of L6 extending into L5 and the top of the white matter (WM) in the contralateral hemisphere (which provide callosal projections). We specifically focused on deep layers of the cortex where there is a transition between L6 and the WM, so that L5 and 6 can be accurately characterized in vivo. Then, we imaged a column 200 μm high (from the WM-L6 border through L6 to L5 – see Supplementary Fig. 12) at 0.2 μm increments in anesthetized mice. The field of view was 40 μm to image the individual beads at high resolution in the lateral direction. Under our experimental conditions, the lateral PSF increased from 0.45 to 0.5 μm and the axial PSF increased from 2 to 2.5 μm from L5 through L6 to WM. This experiment shows that the spherical aberration in the system is fairly low, and degrades the lateral and axial PSF by 10 and 20 % respectively through L5 and 6 to WM. We now mention this finding in the text (lines 507-517).

Comments below are preceded by line numbers:

1) 149-150: "laser blocking filters (BF), and nonlinear imaging filters (F)" not in Fig 1A.

We have corrected Fig. 1a.

2) 152: "mechanical shutter (MS) and a short pass dichroic mirror (DM)." not in Fig 1A.

We have modified Fig. 1a.

3) 281-219: It would be more informative to see the images of ablations at the deeper sites, 450 or 600 μm in Fig 2b.

We have provided images from deeper sites in Supplementary Fig. 6.

4) 208: The apparent point spread function of the soma and basal dendrites in the image of Supplementary Fig 3a does not look very good. Perhaps this is merely because the image in Fig S3a is rendered at very high contrast. One would want to see linear profiles of intensity across the basal dendrites in order to have an accurate idea of the real quality of these images, and to compare it with the point spread functions of Fig 5a. A panel of additional images rendered at lower contrast would also be a worthwhile addition to this manuscript. Perhaps the full z-stack could be provided to the reader for evaluation.

We performed these experiments in awake mice, which are difficult to image stably for long periods of time. We have now provided linear profiles of basal and apical dendrites 50 μm apart from the somata in new Supplementary Fig. 13. The full width half maximum values of these linear profiles are 0.48 and 0.51 μm for apical and basal dendrites, respectively. These values provide the upper bound of lateral resolution and confirm our retrobead measurements (response to reviewer 3, major point above).

5) 230: The spatial resolution in horizontal by vertical pixels as well as the frame rate should be noted. Was it always 256x256?

The spatial resolution was 256x272 pixels for 250x250 μm^2 with 4 Hz frame rate. We have updated this information in the text (lines 470-471).

6) 241-249: Are the responses in Fig 3C selected single trials or averages of multiple trials? If there were selected, how was the selection done? These cells do not look typical of the layers.

These traces are averages of all 10 trials. We did not select trials. We have modified the legend to emphasize this point. We also chose different cells which are typical of layers; e.g., cells with broader OSI for layer 5 and updated Fig. 3c in the main text.

7) 421: Define delta in lines 422-424.

δ is the three-photon cross-section of the fluorophore. We have added this explanation in the text (lines 496-497).

8) 449: Mice have sexes, not "genders"

We have changed the text (line 534).

9) 479: What is meant by "(movement)" in "Neurons were classified as visually (movement) responsive"

We recorded neuronal responses to drifting gratings. This is mentioned in Methods, and we have deleted the term here (line 564).

10) 479: How many different grating directions of movement were presented? 12? How many response trials were recorded for each stimulus direction?

The number of direction was 12 and number of trials was 10 for each direction; this is now clearly stated (lines 564-566).

11) Table 2 should include the fractions as well as the numbers of orientation selective neurons in each cortical layer in each case.

We have added fractions and numbers of orientation selective neurons to Supplementary Table 3 (new version of Table 2).

12) Tables 4, 5, 6: Are the P-values in these tables corrected for multiple comparisons? It looks like they are not, in which case they are not valid. Please correct.

We performed statistical tests with multiple comparison correction and updated Tables 5-7 (new versions of Table 4-6).

Reviewer #4 (Remarks to the Author):

This paper leverages the deep penetration capability of long wavelength 3-photon microscopy for imaging the entire depth of the visual cortex and the subplate neurons. While the technology reported here is similar to that in the previous demonstration of 3P activity imaging, this paper focuses on its application. Specifically, the authors carefully measured the ablative damage threshold in the mouse brain and found that the damage threshold is ~ 10x higher than what is required for 3P imaging. This finding is useful for the field. In addition, there are some interesting results on the orientation tuning of L5/6 and subplate neurons, which shows the promise of 3-photon imaging for neuroscience applications. While I recommend publication of this paper overall, there are a number of details that needs to be corrected. In particular, I found that some of the statements in the paper are either exaggerated or misleading.

We appreciate that the reviewer found our results useful and interesting.

(1) In the introduction, "...Although this study has shown proof-of-concept of functional deep tissue imaging, a major concern is tissue damage. Optimization of laser and microscope parameters through characterization of optical properties of the brain is required to enable valid, reliable and damage-free recording of evoked neuronal responses in deep brain regions in awake mice." This statement appears to imply that the previous work (ref. 14) was not optimized for damage-free recording, but the range of pulse energy and average power for 3P imaging were discussed in ref. 14, and in fact the pulse energy used in the focal plane in ref. 14 is essentially the same as reported in this paper. Some modification of this statement is probably needed to accurately reflect the existing literature.

It was not our intention to imply that 3P imaging in the previous work induced tissue damage. We have modified the statements accordingly and emphasized that detailed analysis of different kinds of tissue damage (optical breakdown, tissue warming), and functional changes in neurons with three-photon microscopy is required (lines 64-67).

(2) "Although there has been some effort to maximize the efficiency of excitation and emission paths of two-photon fluorescence microscopy^{15, 16}, there has been no similar analysis for three-photon." This statement appears to imply that the design for the collection path is different for 2P and 3P imaging. This is not true. The design and analysis are essentially the same for 2P and 3P in terms of signal collection.

We agree with the reviewer that emission pathway is the same for two photon and three photon systems. However, there is no study optimizing both excitation and emission paths in the literature for three-photon microscopy. We have modified this statement (lines 86-87).

(3) "Although this method can label blood vessels and provide information about optical properties of the brain at 1300 nm, it is invasive." It is not clear what the advantages are by using ablation threshold to characterize tissue properties. It is certainly also invasive. While it is valuable to quantify the ablation threshold in brain tissues, I don't think it is a better way to characterize tissue characteristics. In addition, although THG produces stable signal for blood vessels, it is also important to show the signal is not contaminated by other THG sources in Fig.2 (d), since there are structures other than the blood

vessel that can also generate THG signal, some much stronger than blood vessels. Therefore, the use of THG to characterize tissue properties is somewhat problematic due to the non-uniform nature of THG signal.

We agree with the reviewer that the ablation method is invasive. The advantage of ablation is to provide a more precise estimate of attenuation length by quantifying percent of damage with respect to applied laser energy at multiple depths by utilizing only the excitation wavelength in a label-free manner. This method is obviously a better method than labeling blood vessels since multiple depths are studied, and multiple energy levels are applied and quantified in a more controllable way. The precision of this method can be improved by increasing the number of ablation depths so that even small differences in attenuation length of each layer can be detected. Like labeling blood vessels, THG microscopy also utilizes the signal coming from blood vessels but in a label-free way. There is an absorption peak at 430 nm for deoxyhemoglobin¹⁸ so that most of the THG signal is generated from the blood vessels. THG signal in brain is exemplified in previous studies to assist patch-clamping¹⁹ for finding the location of the somata, and axonal tracts in the white matter⁶. No THG study has shown so far any specific/strong signal in the brain rather than blood vessels. Also, previous studies with labeling blood vessels with fluorophores have similar blood vessel picture with THG images as in this study²⁰. As we emphasized in the text, THG imaging is not the most precise method to estimate effective attenuation length so it is an alternative method for laser systems which do not have high power for ablation experiments. We have clarified this in the text (lines 206-208, and lines 212-214).

(4) Fig. 2(d). The THG decay curve looked very segmented or stepwise. Not sure if there is any explanation.

This is due to stepwise change of laser power at every 150 μm depth. We modified legend of Fig. 2 to clarify this point.

(5) Line 99 "...Using half the average power used in a previous design¹⁴," This statement is misleading. While it is true that the power used here is about half, but imaging the hippocampus in ref. 14 requires the penetration of the entire white matter, which necessitates the higher power. Therefore, the comparison in power is not quite valid.

The reviewer is right that we are not imaging hippocampus whereas we imaged the entire cortex and white matter. Our estimation is as follows: Since the effective attenuation length of the cortex is 270 μm and we assume that hippocampus has similar attenuation length, another 100 μm imaging requires $\exp(100/270) = 1.5$ times more power than we used for white matter imaging. Thus, we would need 25 mW average power to image similar hippocampal neurons, which is half of the power used in Ouzounov et al., 2017 (~50 mW). However, we have removed this statement.

(6) The use of a custom objective lens here is not fully justified. It is somewhat puzzling that they have to re-design an objective with all the parameters similar to commercial versions (e.g., Olympus 25x objective, 1.05NA). How does the custom lens compare with commercial objective lenses? Does it really provide significant performance improvement? If not, it will be misleading for the field to believe that custom objective lenses (probably quite expensive) are necessary for 3P imaging.

The objective that we used is indeed available from Olympus (XLPLN25XWMP2), and is designed for longer excitation wavelengths. We modeled the objective in Zemax to continue designing other optics in our system to maximize its performance. We believe that custom-made microscope builders need to design their intermediate and collection lenses to maximize performance. The objective can be also manufactured and compared with Olympus or other manufacturer's objectives but this is not the scope of our study. We also compared the performance of the 3p objective that we used with conventional 2p objective from Olympus to image axonal tracts via THG microscopy with same excitation laser power of 8 mW. Our results show that 3p objective provides two times more signal compared to 2p objective in the same experimental conditions (Supplementary Fig. 3). We have modified the text to clarify this point (lines 162-164). See also response to reviewer 2, point 2.

(7) It is not clear why field curvature is important in the lens design for in vivo brain imaging.

Field curvature is important especially to claim that we are imaging a vertical column in the brain, from individual layers at a time. Otherwise, we may end up recording from several layers at the same time.

(8) Line 165-168. It will be much more convincing to compare the collection efficiency experimentally than using design simulations. There were previous reports indicating that large aperture collection path is not necessary, which contradicts the conclusion here. For example, according to Biomedical Optics Express, Vol. 6, pp. 3113-3127, (2015), even at high scattering, 25 to 30 mm aperture optics are sufficient.

We agree with the reviewer that there are contradictory reports in the literature, based on in vivo brain imaging with different collection schemes. Per the reviewer's suggestion of measuring collection efficiency experimentally, we placed an iris just after the first dichroic mirror before the collection optics and changed its diameter from one to two inches (Supplementary Fig. 5a). Then, we recorded THG signals from the axonal tracts in the white matter in both cases (Supplementary Fig. 5b). We divided both fields of view into five regions and compared average intensity by taking the ratio of the signal acquired with two inch optics and one inch optics. Our results show that there is approximately two times improvement in THG signal with two-inch size optics, which agrees well with our simulation results (Supplementary Fig. 5c). We have added this to the main text (lines 178-182).

(9) The ablation threshold measured here appears to be in good agreement with previous report in PHYSICAL REVIEW B 94, 024113 (2016). Probably useful to cite this paper.

We appreciate the reviewer's comment and cited the paper (line 204-205).

(10) Line 308 "We developed a custom-made pre-chirp system to compensate for pulse broadening in the microscope and were able to reduce the pulse width to 40 fs on the sample. Furthermore, we implemented a custom-made delay line to increase the pulse repetition rate and improve the frame rate." The dispersion compensation and delay line scheme used here are quite standard. Emphasizing "custom-made" is somewhat misleading.

We modified the statements (lines 364-367). However, it should be emphasized that reducing the pulse width on the sample to 40 fs is only possible by developing optimized versions of external compressor, delay line, and all optics in this system.

References

- 1 Stujenske, J. M., Spellman, T. & Gordon, J. A. Modeling the Spatiotemporal Dynamics of Light and Heat Propagation for In Vivo Optogenetics. *Cell Rep* **12**, 525-534, doi:10.1016/j.celrep.2015.06.036 (2015).
- 2 Milburn, T., Saint, D. A. & Chung, S. H. The temperature dependence of conductance of the sodium channel: implications for mechanisms of ion permeation. *Receptors Channels* **3**, 201-211 (1995).
- 3 Kiyatkin, E. A. Brain temperature fluctuations during physiological and pathological conditions. *Eur J Appl Physiol* **101**, 3-17, doi:10.1007/s00421-007-0450-7 (2007).
- 4 Hodgkin, A. L. & Katz, B. The effect of temperature on the electrical activity of the giant axon of the squid. *J Physiol* **109**, 240-249 (1949).
- 5 Aronov, D., Veit, L., Goldberg, J. H. & Fee, M. S. Two distinct modes of forebrain circuit dynamics underlie temporal patterning in the vocalizations of young songbirds. *J Neurosci* **31**, 16353-16368, doi:10.1523/JNEUROSCI.3009-11.2011 (2011).
- 6 Ouzounov, D. G. *et al.* In vivo three-photon imaging of activity of GCaMP6-labeled neurons deep in intact mouse brain. *Nature Methods* **14**, 388+, doi:10.1038/nmeth.4183 (2017).
- 7 Pho, G. N., Goard, M. J., Woodson, J., Crawford, B. & Sur, M. Task-dependent representations of stimulus and choice in mouse parietal cortex. *Nat Commun* **9**, doi:ARTN 259610.1038/s41467-018-05012-y (2018).
- 8 Goard, M. J., Pho, G. N., Woodson, J. & Sur, M. Distinct roles of visual, parietal, and frontal motor cortices in memory-guided sensorimotor decisions. *Elife* **5**, doi:10.7554/eLife.13764 (2016).
- 9 El-Boustani, S. *et al.* Locally coordinated synaptic plasticity of visual cortex neurons in vivo. *Science* **360**, 1349-1354, doi:10.1126/science.aao0862 (2018).
- 10 Mittmann, W. *et al.* Two-photon calcium imaging of evoked activity from L5 somatosensory neurons in vivo. *Nat Neurosci* **14**, 1089-U1195, doi:10.1038/nn.2879 (2011).
- 11 Sun, W. Z., Tan, Z. C., Mensh, B. D. & Ji, N. Thalamus provides layer 4 of primary visual cortex with orientation- and direction-tuned inputs. *Nat Neurosci* **19**, 308+, doi:10.1038/nn.4196 (2016).
- 12 Wang, K. *et al.* Direct wavefront sensing for high-resolution in vivo imaging in scattering tissue. *Nat Commun* **6**, doi:ARTN x10.1038/ncomms8276 (2015).
- 13 Lue, N. *et al.* Tissue refractometry using Hilbert phase microscopy. *Opt Lett* **32**, 3522-3524, doi:Doi 10.1364/Ol.32.003522 (2007).
- 14 Rappaz, B. *et al.* Measurement of the integral refractive index and dynamic cell morphometry of living cells with digital holographic microscopy. *Opt Express* **13**, 9361-9373, doi:Doi 10.1364/Opex.13.009361 (2005).
- 15 Vogelstein, J. T. *et al.* Fast Nonnegative Deconvolution for Spike Train Inference From Population Calcium Imaging. *J Neurophysiol* **104**, 3691-3704, doi:10.1152/jn.01073.2009 (2010).
- 16 Niell, C. M. & Stryker, M. P. Highly selective receptive fields in mouse visual cortex. *J Neurosci* **28**, 7520-7536, doi:10.1523/JNEUROSCI.0623-08.2008 (2008).

- 17 Keller, G. B., Bonhoeffer, T. & Hubener, M. Sensorimotor Mismatch Signals in Primary Visual Cortex of the Behaving Mouse. *Neuron* **74**, 809-815, doi:10.1016/j.neuron.2012.03.040 (2012).
- 18 Chang, C. F., Yu, C. H. & Sun, C. K. Multi-photon resonance enhancement of third harmonic generation in human oxyhemoglobin and deoxyhemoglobin. *J Biophotonics* **3**, 678-685, doi:10.1002/jbio.201000045 (2010).
- 19 Witte, S. *et al.* Label-free live brain imaging and targeted patching with third-harmonic generation microscopy. *Proc Natl Acad Sci U S A* **108**, 5970-5975, doi:10.1073/pnas.1018743108 (2011).
- 20 Kobat, D., Horton, N. G. & Xu, C. In vivo two-photon microscopy to 1.6-mm depth in mouse cortex. *J Biomed Opt* **16**, doi:Artn 10601410.1117/1.3646209 (2011).

REVIEWERS' COMMENTS:

Reviewer #1:

[In remarks to the editor, said she/he is happy to recommend publication as all issues were addressed.]

Reviewer #4 (Remarks to the Author):

The revised manuscript is significantly improved from the original submission. The inclusion of physiological response as a function of pulse energy is valuable information for the applications of 3P imaging. I would recommend acceptance of this paper.

Two minor issues:

(1) The paper states that GCaMP saturation happens at pulse energy > 5 nJ. It is not clear how the authors arrived at this conclusion. At pulse energy between 2 to 5 nJ, the authors show that the response intensity is reduced, but this could simply be caused by GCaMP saturation, instead of perturbation of physiological function. Depending on the exact 3P cross section values, it is entirely possible that GCaMP saturation happens at pulse energy between 2 to 5 nJ. More careful discussion might be needed here.

(2) I am still not convinced about the merits of the custom objective lens. The long wavelength multiphoton objective lens from Olympus also has transmission of $\sim 80\%$ at 1300 nm, which is similar to the custom lens.

Response to Reviewer 4

We thank the reviewer 4 for his/her last comments about our paper. Our point-by-point response is provided below.

Reviewer #4 (Remarks to the Author):

The revised manuscript is significantly improved from the original submission. The inclusion of physiological response as a function of pulse energy is valuable information for the applications of 3P imaging. I would recommend acceptance of this paper.

Two minor issues:

(1) The paper states that GCaMP saturation happens at pulse energy > 5 nJ. It is not clear how the authors arrived at this conclusion. At pulse energy between 2 to 5 nJ, the authors show that the response intensity is reduced, but this could simply be caused by GCaMP saturation, instead of perturbation of physiological function. Depending on the exact 3P cross section values, it is entirely possible that GCaMP saturation happens at pulse energy between 2 to 5 nJ. More careful discussion might be needed here.

The probability of fluorophore excitation per laser pulse (Pr_{pulse}) should be less than 0.1 ($Pr_{pulse} < 0.1$) to avoid excitation saturation and point spread function broadening. Pr_{pulse} can be calculated as follows:

$$Pr_{pulse} = \frac{\delta_a P_0^3}{\tau_p^2 f_p^3} \left(\frac{NA^2}{2hc\lambda} \right)^3 = \frac{\delta_a E_0^3}{\tau_p^2} \left(\frac{NA^2}{2hc\lambda} \right)^3$$

Where three-photon cross section of the fluorophore (δ_a) is $1-2 \times 10^{-82} \text{ cm}^6(\text{s/photon})^2$ [1], average laser power (P_0) is 4 mW, pulse energy (E_0) is 5 nJ, pulse width (τ_p) is 40 fs, pulse repetition rate (f_p) is 800 kHz, numerical aperture (NA) is 0.9, h is Planck's constant, c is speed of light, and wavelength of the laser (λ) is 1300 nm. With these conditions, Pr_{pulse} varies between 0.03-0.06. Therefore, we can conclude that we can avoid saturation of GCaMP6s with pulse energies less than 5 nJ. This calculation agrees well with our experimental observation that the saturation of GCaMP6s is initiated with pulse energies > 5 nJ (Supplementary Fig. 7, the third column) where whole field of view is blurred. We added this calculation and explanation of fluorophore saturation in the discussion part.

(2) I am still not convinced about the merits of the custom objective lens. The long wavelength multiphoton objective lens from Olympus also has transmission of ~ 80% at 1300 nm, which is similar to the custom lens.

We would like to reemphasize here that we modeled the objective lens from Olympus in order to design and manufacture excitation and emission path optical elements. To compare the performance of the commercial lens and custom-made lens is not scope of our work.

1. Xu, C., et al., *Multiphoton fluorescence excitation: New spectral windows for biological nonlinear microscopy*. Proceedings of the National Academy of Sciences of the United States of America, 1996. **93**(20): p. 10763-10768.